# Zinc isotopes from archaeological bones provide reliable trophic level information for marine mammals

Jeremy McCormack [1✉], Paul Szpak[2], Nicolas Bourgon[1,3], Michael Richards [4], Corrie Hyland[2], Pauline Méjean[5], Jean-Jacques Hublin [1] & Klervia Jaouen[1,5]

In marine ecology, dietary interpretations of faunal assemblages often rely on nitrogen isotopes as the main or only applicable trophic level tracer. We investigate the geographic variability and trophic level isotopic discrimination factors of bone zinc $^{66}Zn/^{64}Zn$ ratios ($\delta^{66}Zn$ value) and compared it to collagen nitrogen and carbon stable isotope ($\delta^{15}N$ and $\delta^{13}C$) values. Focusing on ringed seals (*Pusa hispida*) and polar bears (*Ursus maritimus*) from multiple Arctic archaeological sites, we investigate trophic interactions between predator and prey over a broad geographic area. All proxies show variability among sites, influenced by the regional food web baselines. However, $\delta^{66}Zn$ shows a significantly higher homogeneity among different sites. We observe a clear trophic spacing for $\delta^{15}N$ and $\delta^{66}Zn$ values in all locations, yet $\delta^{66}Zn$ analysis allows a more direct dietary comparability between spatially and temporally distinct locations than what is possible by $\delta^{15}N$ and $\delta^{13}C$ analysis alone. When combining all three proxies, a more detailed and refined dietary analysis is possible.

[1] Department of Human Evolution, Max Planck Institute for Evolutionary Anthropology, Leipzig, Germany. [2] Department of Anthropology, Trent University, Peterborough, ON, Canada. [3] Applied and Analytical Paleontology, Institute for Geosciences, Johannes Gutenberg-University Mainz, Mainz, Germany. [4] Department of Archaeology, Simon Fraser University, Burnaby, BC, Canada. [5] Géosciences Environnement Toulouse, UMR 5563, CNRS, Observatoire Midi Pyrénées, Toulouse, France. ✉email: jeremy_mccormack@eva.mpg.de

In ecology, archaeology and palaeontology, accurately reconstructing trophic levels can be challenging. Among others, these reconstructions are required for effective management and conservation strategies[1], understanding changing predator-prey and foraging ecology related to climate change[2] and studying long-term environmental changes through the analyses of modern to fossil faunal assemblages (and their bone collagen stable isotope values)[3,4]. Stable isotope analysis is an effective tool for analysing marine food webs, complementary to and often more reliable than non-stable isotope approaches[5]. Bone collagen and soft tissue bulk $\delta^{15}N$ and $\delta^{13}C$ values are the traditional geochemical proxies used for dietary and trophic level reconstructions[6–8]. These isotope analyses can now be conducted on single collagen amino acids and provide more refined information on the trophic level of animals (e.g., corrected for baseline variability)[9,10], but these studies are still time consuming and expensive.

Only recently, studies of the bone's mineral phase (bioapatite) non-traditional isotope systems such as calcium, magnesium and zinc (Zn) have shown potential as (palaeo)dietary and trophic level proxies in the terrestrial[11–13] and marine realm[14,15]. Element and isotope ratios observed in a diagenetically more resistant mineral phase (e.g., enamel) can preserve dietary information beyond the scope of application of collagen, as recently demonstrated for $^{66}Zn/^{64}Zn$ ratios (expressed as the $\delta^{66}Zn$ value)[16]. In addition, even when collagen is well preserved, combining traditional collagen $\delta^{15}N$ and $\delta^{13}C$ analyses with Zn isotope analyses of bioapatite may provide complementary dietary information as proven by the lack of correlation of those tracers within individuals of the same species, implying independent controlling mechanisms[17]. Therefore, analysing $\delta^{66}Zn$ alongside traditional $\delta^{15}N$ analyses may help verify $\delta^{15}N$ results and provide a much-needed $\delta^{15}N$-independent additional geochemical trophic level and dietary indicator for ecological and archaeological studies.

Nitrogen-15 becomes relatively enriched in the tissues of aquatic consumers with successive trophic level[18]. The $\delta^{13}C$ values behave much more conservatively with trophic level, increasing typically by <1‰ with trophic level for most tissues, compared to on average 3.4‰ higher $\delta^{15}N$ values between a predator and its prey[19,20]. Carbon isotopes are therefore more commonly used to infer the source(s) of primary production at the base of the food web. For Zn, studies have shown a distinct $^{66}Zn$ depletion in carnivore bioapatite relative to that of herbivores[13,16]. As muscles and most organs are typically $^{66}Zn$ depleted relative to the animal's diet and its bulk body $\delta^{66}Zn$ composition[21–23], bones of carnivores (and their bulk body composition) have lower $\delta^{66}Zn$ values than their prey's. Diet thus exerts control on the $\delta^{66}Zn$ values of soft tissue and bioapatite. Most non-diet related factors, such as sex and age of an animal, have so far shown no effect on the isotope values and relative isotopic variability in body tissues[23,24]. Provenance, on the other hand, appears to affect the Zn isotope compositions of terrestrial vertebrates, though it remains unclear to what extent[13,16,25]. Provenance is known to play an important role for bulk $\delta^{15}N$ and $\delta^{13}C$ studies in marine food webs[26] but has until now not been studied for $\delta^{66}Zn$.

In the marine realm, the $\delta^{15}N$ and $\delta^{13}C$ values of particulate organic matter (POM), consisting of phytoplankton, bacteria, microzooplankton and detritus, show a substantial spatial variation within and among ocean basins[5,26–28]. Variation in the isotopic composition at the base of the marine food web is passed along to higher trophic levels. Particularly for bone collagen, with its long turnover time[29], trophic level reconstructions can be compromised when animals frequently migrate between areas of differing food web baseline isotopic composition, or when comparing animals from spatially and temporally distinct locations.

Here, we aim at exploring Zn isotopes as a reliable tracer of marine trophic levels. To do so, we investigate combined bone $\delta^{66}Zn$, $\delta^{15}N$ and $\delta^{13}C$ from the same species across 13 locations (17 sites) in a large geographic area stretching across the Arctic from the Hudson Strait in the east to the Bering Strait in the west. We include 5 locations and two single samples with already published $\delta^{15}N$ and $\delta^{13}C$ values[4,15,30,31], as well as one site with already published $\delta^{66}Zn$ values[15]. For this study, we analysed 167 archaeological bones, concentrating on ringed seals Pusa hispida (Phoca hispida) and polar bears (Ursus maritimus). Focusing on these species allows us to investigate $\delta^{66}Zn$ trophic level isotope discrimination factors between predator and prey geographically. Both species have a circumpolar Arctic distribution and are abundant throughout the Arctic today[32,33]. Particularly, P. hispida remains are frequently found in archaeological assemblages with a large temporal and spatial range[30,34,35].

As we focus here on archaeological material, we have no direct food web baseline information for the isotope systems analysed herein. Therefore, we take advantage of multiple studies documenting P. hispida as the main prey species of U. maritimus and subsequently both species being close to one trophic level apart[8,36–39] (Supplementary Note 1.1). We can therefore use these taxa to estimate species to species relative trophic level variability across the here studied sites. Studying the isotopic composition of high trophic level predators such as U. maritimus and P. hispida also has the advantage of their tissues' isotopic composition dampening the effects of short-term environmental variation and integrating multiple food web channels. This effectively leads to less isotopic variability and "noise" in the animal's tissues compared to those of lower trophic levels[30]. Therefore, these two species are prime targets to investigate the geographical variability of dietary proxies and trophic level isotopic discrimination factors. Anticipating an advantage of combining $\delta^{66}Zn$ with $\delta^{15}N$ and $\delta^{13}C$ values, we also test in a series of bone dissolution experiments whether collagen-bound Zn influences bone $\delta^{66}Zn$ values, which would preclude combining collagen extraction protocols with $\delta^{66}Zn$ analysis of the mineral phase without resampling material.

## Isotopic context

Marine phytoplankton from high latitudes shows particularly high variability in $\delta^{13}C$ values[40]. Colder surface water temperatures lead to increasing aqueous $CO_2$ content, and therefore a net transfer of isotopically light $CO_2$ to the ocean and a depletion of $^{13}C$ in the surface water[41]. Other factors influencing spatial POM $\delta^{13}C$ values include phytoplankton growth rates, cell size and cell lipid content (see ref. [5,40] and ref. therein). Additional spatial variability may arise from the relative contribution of sea ice POM (sympagic-POM) and open water phytoplankton (pelagic-POM) to a food web. Coeval pelagic- and sympagic-POM have differing $\delta^{13}C$ values, with the former being $^{13}C$ depleted relative to the latter by 2–10‰[37,42–44]. Subtle shifts in Arctic consumers' $\delta^{13}C$ values for a specific area over time may occur with large-scale shifts in the relative importance of sympagic versus pelagic production related to changes in sea ice extent[4,30].

A high variability in modern baseline $\delta^{15}N$ and $\delta^{13}C$ values is documented by the isotopic composition of POM, zooplankton, higher trophic level consumers, as well as filter feeders across the Arctic[6,45–50]. Pomerleau et al.[48] documented a significant spatial variability in zooplankton $\delta^{15}N$ values among the Labrador Sea, Baffin Bay and the Canadian Arctic Archipelago (CAA), but not for $\delta^{13}C$ among these areas. Subsequent studies documented a higher variability for $\delta^{13}C$ of POM, zooplankton and high trophic level consumers among and within these areas as well[49,50]. In addition, a pronounced west-east $^{13}C$ depletion was observed

throughout consumers from the Bering Sea (Bering Strait) through the Chukchi Sea to the Beaufort Sea[6,45,46,49]. A similar west-east trend was also found for sedimentary organic carbon accumulated along the Beaufort Shelf[40]. This eastward $^{13}C$ depletion trend reaches its maximum in the south-eastern Beaufort Sea. Terrestrial organic matter derived from the Mackenzie River has $\delta^{13}C$ values of ~−26 to −27‰ and dominates over autochthonous organic matter in the delta and at least parts of the Beaufort shelf[51]. Terrigenous $^{13}C$ depleted carbon is also thought to play an important role for some animals (gammarid amphipods) of the Mackenzie shelf's food web[52]. A similar variability in $\delta^{15}N$ values between the Bering Sea, Chukchi Sea and Beaufort Sea is absent within animals of higher trophic levels[6,53]. However, geographic variations in $\delta^{15}N$ values within these water bodies were observed for different zooplankton species[6,46,49]. Parson et al.[52] explained high $\delta^{15}N$ values in POM of the Mackenzie estuary instead of a low terrigenous signal as a potential bacterial recycling of nitrogen.

The eastward decrease of baseline $\delta^{13}C$ values does not seem to continue into the CAA[43,48], but significantly lower $\delta^{13}C$ values have been reported in consumer tissues close to the Canadian mainland and in semi-enclosed basins[54]. In accordance, De La Vega et al.[50] observed higher baseline $\delta^{13}C$ values in inflow shelves connected to the Atlantic or Pacific Oceans (Barents Sea, Chukchi Sea) and the North Water Polynya (Northern Baffin Bay) compared to lower baseline values in the more freshwater-influenced Arctic shelves (Beaufort Sea, CAA, Hudson Bay). Lower baseline values for carbon in the more terrestrial-influenced areas are likely a result of terrigenous input and lower phytoplankton productivity. Higher stratification caused by inflowing freshwater hampers phytoplankton productivity on the interior shelves[55]. Indeed, Bering and Chukchi Sea annual primary production rates greatly exceed those of the Beaufort Sea[6].

Zn isotopes are increasingly being used as tracers for past marine hydrochemistry[56,57] and culture experiments have investigated Zn isotope fractionation in different planktonic species[58,59]. Still, there is hardly any data on the Zn isotope composition of natural marine planktonic organisms[60,61]. Indeed, data on food web baseline $\delta^{66}Zn$ values and variability is non-existent for both the marine and continental realms. Because of biological uptake, dissolved Zn concentrations are highly depleted in marine surface waters, often much <1 nmol kg$^{-1}$[62,63] and most oceans show a nutrient-like vertical distribution of dissolved Zn concentrations closely correlating with silicate concentrations[64].

The isotopic composition of dissolved Zn below 500 m seems to be globally homogenous with values close to +0.5‰, despite variable Zn concentrations[65,66]. The bulk isotopic composition of dissolved marine Zn is enriched in $^{66}Zn$ relative to its major inputs from rivers and aeolian dust, which centre on the global crustal average of +0.3‰[67].

Although most studies on cultured phytoplankton demonstrate a preferential uptake of light Zn into the cell relative to the bulk growth medium[58,59], Atlantic and Pacific vertical dissolved Zn isotope profiles generally show lower $\delta^{66}Zn$ values in surface waters compared to that of the deep water[62,64,66,68,69]. These studies demonstrate that the isotopic composition of Zn is most variable within the surface water (<500 m), often with higher values in the uppermost surface (<20 m). Surface water dissolved Zn isotope ratios vary across a North Atlantic transect from −1.1 to +0.9‰[62] and across a North Pacific transect between −0.9 and +0.2‰[69]. Individual and combined mechanisms discussed to be responsible for this surface water $\delta^{66}Zn$ variability include external inputs from rivers and aerosols[67,69], scavenging of heavy Zn onto sinking organic matter[64] and biological uptake and shallow remineralisation[70].

We are unaware of any $\delta^{66}Zn$ data from dissolved Zn in the Arctic. However, a recent study on Western Arctic dissolved Zn concentrations highlighted a deviation of Zn concentration vertical profiles from the nutrient-type Zn profiles observed in the Atlantic and Pacific[71]. These authors documented higher than global average surface Zn concentrations (~1.1 nmol kg$^{-1}$) with a maximum concentration at 200 m and uniformly lower concentrations in the deep water. Jensen et al.[71] hypothesises that Western Arctic surface water dissolved Zn originates primarily from incoming Pacific waters that are modified by shelf sediment fluxes from remineralised Zn-rich phytoplankton.

## Results

**Bone $\delta^{13}C$, $\delta^{15}N$ and $\delta^{66}Zn$ values.** Bone collagen $\delta^{13}C$, $\delta^{15}N$ and bone $\delta^{66}Zn$ values of *P. hispida*, *U. maritimus*, harp seal (*Pagophilus groenlandicus*) and beluga whale (*Delphinapterus leucas*) are reported in Table 1 and Supplementary Data 1. All collagen samples had yields and elemental (wt% C, wt% N, C: N$_{atomic}$) compositions characteristic of samples with isotopic compositions not altered by contaminant or degradation in the burial environment[72,73] (Supplementary Data 1). Likewise, $\delta^{66}Zn$ values do not indicate a modification due to diagenesis or contamination for the majority of samples, but we cannot exclude it

**Table 1 Isotopic range ($\delta^{13}C$, $\delta^{15}N$, $\delta^{66}Zn$) for all bone samples discussed in this study for which all three elements were analysed.**

| Species | n | | $\delta^{13}C$ (‰, VPDB) | $\delta^{15}N$ (‰, AIR) | $\delta^{66}Zn$ (‰, JMC Lyon) | [Zn] (µg/g) | TL after[8,37,80,103] |
|---|---|---|---|---|---|---|---|
| *U. maritimus* | 47 | max. | −12.49 | +24.41 | +0.73 | 901 | 5.1–5.5 |
| | | min. | −15.04 | +16.78 | −0.06 | 80 | |
| | | mean | −13.68 | +21.75 | +0.17 | 276 | |
| | | SD | 0.65 | 1.72 | 0.16 | 155 | |
| *P. hispida* | 104 | max. | −12.00 | +19.56 | +0.76 | 878 | 3.8–4.6 |
| | | min. | −16.95 | +14.25 | +0.23 | 79 | |
| | | mean | −14.48 | +17.22 | +0.49 | 167 | |
| | | SD | 1.13 | 1.15 | 0.10 | 105 | |
| *P. groenlandicus* | 11 | max. | −13.60 | +17.45 | +0.66 | 273 | 3.8–3.9 |
| | | min. | −14.61 | +13.84 | +0.22 | 82 | |
| | | mean | −14.18 | +15.47 | +0.46 | 135 | |
| | | SD | 0.35 | 1.26 | 0.14 | 57 | |
| *D. leucas* | 2 | max. | −11.24 | +18.18 | +0.67 | 1025 | 3.9–4.4 |
| | | min. | −13.21 | +17.75 | +0.65 | 381 | |
| | | mean | −12.23 | +17.97 | +0.66 | 703 | |

*Max.* maximum value, *min.* minimum value, *SD* standard deviation, *n* number of individuals/bone samples, [Zn] Zn concentration, *TL* trophic level range estimates from the literature with citations.

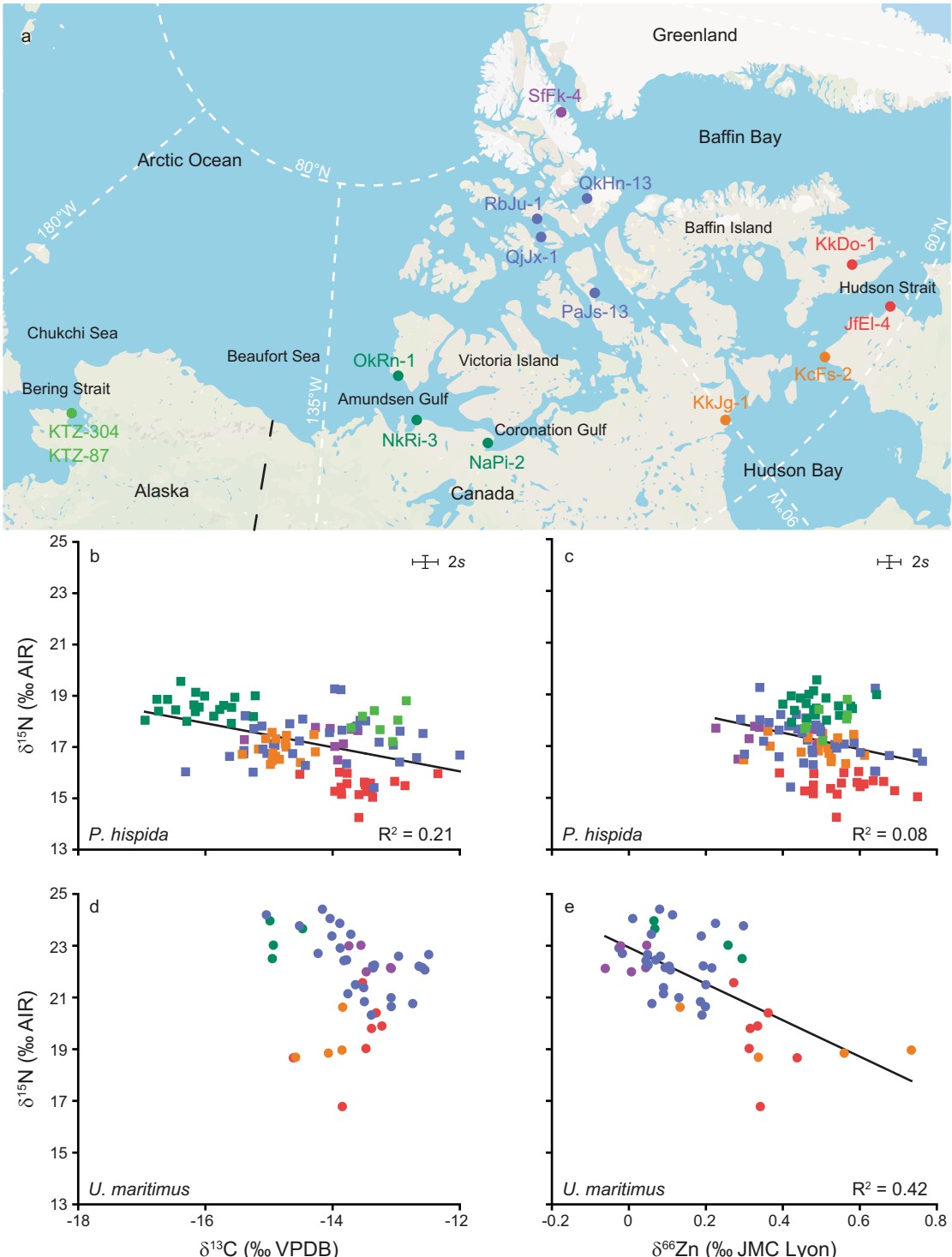

**Fig. 1 Isotopic composition of *P. hispida* and *U. maritimus* bone samples from Arctic archaeological sites.** *Pusa hispida* (squares, $n = 104$) and *U. maritimus* (dots, $n = 47$) bone samples are colour coded as geographic groups. **a** Schematic map indicating the archaeological sites analysed and geographic colour coding: Light green for the Bering Strait; dark green for the Amundsen and Coronation Gulf; blue for the CAA; orange for the Hudson Bay; purple for the North Water Polynya; and red for sites influenced by the Labrador Sea in the Hudson Strait and Frobisher Bay. **b** $\delta^{15}N$ *versus* $\delta^{13}C$ plot for *P. hispida* samples (*p*-value < 0.05; $R^2 = 0.21$; $n = 104$). **c** $\delta^{15}N$ *versus* $\delta^{66}Zn$ plot for *P. hispida* samples (*p*-value < 0.05; $R^2 = 0.08$; $n = 104$). **d** $\delta^{15}N$ *versus* $\delta^{13}C$ plot for *U. maritimus* samples (no correlation, *p*-value > 0.05; $n = 47$). **e** $\delta^{15}N$ *versus* $\delta^{66}Zn$ plot for *U. maritimus* samples (*p*-value < 0.05; $R^2 = 0.42$; $n = 47$). We included already published $\delta^{15}N$ and $\delta^{13}C$ values[4,15,30,31] and already published $\delta^{66}Zn$ values from QjJx-1[15]. The map is redrawn and modified using Adobe Illustrator CS6 after www.google.com/maps. Error bars represent the measurement uncertainty.

as a possibility for outlier values (Supplementary Discussion 3.1, Supplementary Fig. 1). Within a site, we observe typically higher bulk collagen $\delta^{15}N$ values and lower bone mineral $\delta^{66}Zn$ values in *U. maritimus* relative to *P. hispida* and other prey species (Table 1, Fig. 1). Our results from applying different dissolution methods to reference materials and bone samples also indicate that the presence of collagen-bound Zn, and thereby collagen preservation, has no effect on the mineral phase $\delta^{66}Zn$ values (Supplementary Methods 2.1). Mineral phase $\delta^{66}Zn$ analyses can thus be coupled with collagen extraction protocols, provided precautions are taken to avoid Zn contamination (Supplementary Figs. 2–3, Supplementary Data 2, Supplementary Discussion 3.2).

**Statistical investigation of isotope values.** Statistically significant differences between *P. hispida* populations was determined through ANOVA for $\delta^{13}C$ ($F_{(12, 91)} = 24.4$, p-value $< 2e^{-16}$), and $\delta^{66}Zn$ values ($F_{(12, 91)} = 5.867$, p-value $1.93e^{-07}$), and through Welch ANOVA for $\delta^{15}N$ ($F_{(12, 32.2)} = 71.8$, p-value $1.00e^{-19}$). Post hoc Tukey pair-wise comparisons draw out the populations from Little Cornwallis (QjJx-1) and the North shore of Devon Island (QkHn-13) both part of the CAA, as well as eastern Ellesmere Island (near Skraeling Island, SfFk-4), linked to the North Water Polynya, as distinct from some of the other sites in regard to their $\delta^{66}Zn$ values (Supplementary Figs. 4–5, Supplementary Data 3). Every other site, regardless of their broad geographic group, are not significantly different from one another.

Results for pair-wise comparisons of sites' $\delta^{13}C$ and $\delta^{15}N$ *P. hispida* values show a higher degree of heterogeneity (Supplementary Figs. 6–7, Supplementary Data 3). However, most of the differences can be linked to geographic groups. Sites from the CAA are being drawn out as different in their $\delta^{13}C$ and $\delta^{15}N$ values to most of the other sites. The western sites of the Amundsen and Coronation Gulf, as well as the Bering Strait, differ in their $\delta^{15}N$ values, but not for $\delta^{13}C$ values. Finally, $\delta^{13}C$

and $\delta^{15}N$ values from the Eastern sites of the Hudson Bay and the Labrador Sea are identified as significantly different than those of western sites.

Levene's tests for equal variance show that $\delta^{66}Zn$ values are more homogeneous between *P. hispida* and *U. maritimus* ($F_{(1, 125)} = 3.43$, $p = 0.27$) and across sites ($F_{(8, 118)} = 1.72$, $p = 0.40$) than $\delta^{15}N$ values (respectively $F_{(1, 125)} = 6.95$, $p = 0.04$; and $F_{(1, 118)} = 2.62$, $p = 0.05$).

## Discussion

The maximum variability for inter-site mean *P. hispida* bone $\delta^{15}N$ and $\delta^{13}C$ values (3.55 and 3.40‰) exceeds the maximum intra-site (1.77 and 2.67‰, Fig. 1) and typical trophic level variability in $\delta^{15}N$ between predator and prey (e.g., +3.4 to +3.8‰)[8,18,19]. The QjJx-1 site on Little Cornwallis Island is a notable exception with very high on-site *P. hispida* bone collagen $\delta^{15}N$ variability (3.85‰)[15]. Post hoc Games-Howell and Tukey pair-wise comparisons demonstrate a large heterogeneity in $\delta^{15}N$ and $\delta^{13}C$ values between archaeological populations (Supplementary Figs. 6–7). Isotopic heterogeneity between populations is related to geographic location resulting in $\delta^{15}N$ and $\delta^{13}C$ values from populations of different regions plotting in distinct groups on a $\delta^{15}N$ versus $\delta^{13}C$ plot (Fig. 1b). Based on site proximity and sample isotopic composition, we grouped sites from the Bering/ Chukchi Sea, Amundsen and Coronation Gulf, CAA, North Water Polynya, Hudson Bay, and sites influenced by the Labrador Sea (Hudson Strait and Frobisher Bay, Fig. 1a). The $\delta^{15}N$ and $\delta^{13}C$ variability between the archaeological sites is in good agreement in both spacing and amplitude with modern geographical variations observed from zooplankton[28,45,46] and higher consumer soft tissue[6,49] including *P. hispida*[37,53,54,74–76] and *U. maritimus*[33]. While dietary differences between populations may have contributed to the geographic spacing of $\delta^{15}N$ and $\delta^{13}C$ values, integration of regional baseline isotopic patterns is likely the main factor controlling the observed inter-site isotopic

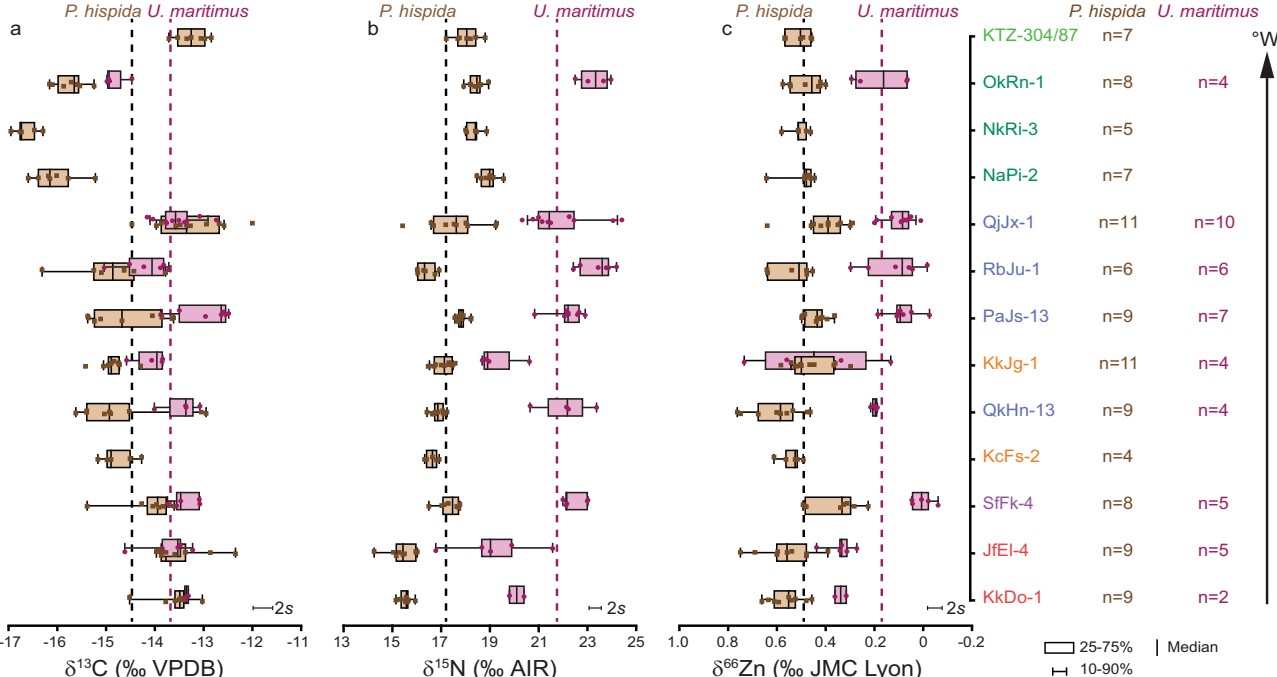

**Fig. 2 Range of $\delta^{13}C$, $\delta^{15}N$ and $\delta^{66}Zn$ values for *P. hispida* and *U. maritimus* bones.** Range of $\delta^{13}C$ (**a**), $\delta^{15}N$ (**b**) and $\delta^{66}Zn$ (**c**) values for *P. hispida* and *U. maritimus* bones for all locations. Site names are colour coded following Fig. 1. Dashed lines represent mean values when including all sites. We included already published $\delta^{15}N$ and $\delta^{13}C$ values from sites RbJu-1, PaJs-13, QkHn-13, KTZ and QjJx-1 sites[4,15,30,31] and already published $\delta^{66}Zn$ values from QjJx-1[15]. Error bars represent the measurement uncertainty.

variability in *P. hispida* and *U. maritimus* collagen (Supplementary Discussion 3.3).

In contrast to *P. hispida* bone $\delta^{15}N$ and $\delta^{13}C$ values, the highest variability for mean $\delta^{66}Zn$ values between sites (0.23‰) does not exceed the maximum variability observed within a single site (0.36‰). In addition, mean $\delta^{66}Zn$ values between sites never exceed the variability between mean *U. maritimus* and *P. hispida* bone $\delta^{66}Zn$ values ($\Delta^{66}Zn_{U.\ maritimus\ -\ P.\ hispida} = -0.32$‰). While ANOVA analysis of *P. hispida* $\delta^{66}Zn$ values did reveal statistically significant differences between *P. hispida* populations, post hoc pair-wise comparison tests show considerably more homogeneity in $\delta^{66}Zn$ values than for $\delta^{15}N$ and $\delta^{13}C$ values (Supplementary Figs. 4–7). Similarly, Levene's tests for equal variance show that for all sites, *P. hispida* and *U. maritimus* $\delta^{66}Zn$ values have an equal variance, whereas $\delta^{15}N$ values are more heterogeneous. The low geographical $\delta^{66}Zn$ variability in *P. hispida* and *U. maritimus* bones implies that Arctic food web baseline and/or low trophic level consumer $\delta^{66}Zn$ values are more homogenous than for $\delta^{15}N$ and $\delta^{13}C$ values. This is remarkable considering the large surface water's isotopic variability observed for dissolved Zn across the Atlantic and Pacific of $-1.1$ to $+0.9$‰ and $-0.9$ to $+0.2$‰, respectively[62,69].

Based on post hoc Tukey pair-wise comparison, $\delta^{66}Zn$ values from *P. hispida* populations from the sites QjJx-1 (Little Cornwallis Island), QkHn-13 (Devon Island) and SfFk-4 (eastern Ellesmere Island) were identified as statistically different from other populations (Fig. 2, Supplementary Fig. 4). Differences in the dietary Zn resources of these populations relative to others may have caused these statistical anomalies. Alternatively, they may reflect true variability in the $\delta^{66}Zn$ regional food web baselines. QjJx-1 and QkHn-13 are located within the CAA. The CAA is composed of multiple channels and interconnected basins, in which water mass modification and transport are governed by its complex topography and shelf exchange processes[77]. Within this setting, baseline $\delta^{66}Zn$ values are perhaps more variable on a regional scale than for the rest of the Arctic. For the SfFk-4 site, we observe in all three species analysed lower mean $\delta^{66}Zn$ values compared to other sites indicating a regionally lower baseline $\delta^{66}Zn$ value (Figs. 2, 3). The SfFk-4 site is located at the biologically highly productive[78] northern edge of the North Water Polynya, a region in which the reduced ice-cover or ice-free conditions influence biological processes (e.g., by upwelling, increased nutrient renewal)[79], which in turn may modify the $\delta^{66}Zn$ baseline.

Both $\delta^{66}Zn$ and $\delta^{15}N$ values are controlled by diet but show a better correlation for *U. maritimus* samples across all sites than for *P. hispida* (Fig. 1c, e), perhaps related to the more specialised diet of *U. maritimus*[38,39]. As with $\delta^{15}N$ values, bone $\delta^{66}Zn$ values clearly demonstrate a trophic spacing between *U. maritimus* and *P. hispida* in all locations analysed (Fig. 2). The KkJg-1 site in Hudson Bay is an exception with two *U. maritimus* samples showing anomalously high $\delta^{66}Zn$ values which may relate to non-dietary factors such as contamination, misidentification, diagenesis, or physiological effects (Supplementary Discussion 3.1). Even when including the KkJg-1 site, Levene's tests for equal variance demonstrate an equal variance between *P. hispida* and *U. maritimus* $\delta^{66}Zn$ values, whereas their $\delta^{15}N$ values demonstrate heterogeneity. Because $\delta^{66}Zn$ is more homogenous in its value for a specific taxon, and possibly diet, $\delta^{66}Zn$ may more reliably reflect trophic levels than bulk $\delta^{15}N$ values, when investigating multiple species across multiple sites, proving a better inter-site comparability.

*Ursus maritimus* bone $\delta^{66}Zn$ values are on average 0.32‰ lower than those of *P. hispida* (mean $\Delta^{66}Zn_{U.\ maritimus\ -\ P.\ hispida} = -0.32$‰). Because *P. hispida* is typically the primary prey species of *U. maritimus* for most locations today[38,39], we predict this $\Delta^{66}Zn_{U.\ maritimus\ -\ P.\ hispida}$ value to be close to the Zn bone trophic level discrimination factor between a carnivore and its prey, when soft tissue is consumed. Previously, estimations of trophic discrimination factors between bioapatite of terrestrial mammalian carnivores and herbivores were between $-0.6$ and $-0.4$‰, respectively for the Tam Hay Marklot (THM) cave[16] and the modern Koobi Fora region[13]. These studies, however, had a lower sample size and compared multiple carnivores and herbivores with varying diets. Predicted bone $\delta^{66}Zn$ trophic level discrimination factors are between $-0.36$ and $-0.38$‰ when calculated using individual $\delta^{15}N$ trophic levels[8] from all marine mammal taxa with available $\delta^{66}Zn$ data (Supplementary Discussion 3.4, Supplementary Tables 1–2, Supplementary Fig. 8). However, these estimations are oversimplified, not considering population-specific dietary differences, location-specific baseline variations and organism-specific trophic and tissue-type enrichment factors. We cannot exclude different $\delta^{66}Zn$ trophic level discrimination factors between tissues of *P. hispida* relative to their prey (which could not be analysed herein). The $\delta^{66}Zn$ values of different tissues vary within an organism[21–23]. It is therefore possible that when different tissues are consumed (e.g., consumption of soft tissue only *versus* consumption of the whole organisms), $\delta^{66}Zn$ trophic level discrimination factors might vary as well. Nevertheless, bone $\delta^{66}Zn$ trophic level discrimination factors calculated from $\delta^{15}N$ trophic levels[8] are close to the mean $\Delta^{66}Zn_{U.\ maritimus\ -\ P.\ hispida}$ value and should at least approximate true discrimination factors between bones of a carnivore and its prey.

Particularly for archaeological material, assigning a relative trophic level to multiple species when utilising $\delta^{15}N$ values alone can be challenging, as shown by the large differences in mean $\delta^{15}N$ trophic discrimination factors between *U. maritimus* and *P. hispida* ($\Delta^{15}N_{U.\ maritimus\ -\ P.\ hispida}$) for individual sites ($+2.2$ to $+7.0$ ‰, Fig. 2b). Besides locally differing diet, $\Delta^{15}N_{U.\ maritimus\ -\ P.\ hispida}$ variability (and $\Delta^{66}Zn_{U.\ maritimus\ -\ P.\ hispida}$) may be influenced by physiological effects or unknown archaeological assemblage effects related to human hunting and/or scavenging. Relative differences in the consumption of higher and lower trophic level prey alone are unlikely to explain the $\Delta^{15}N_{U.\ maritimus\ -\ P.\ hispida}$ variability. As *P. hispida* bones are the most abundant fauna remains in all archaeological sites analysed herein[30,34,35], it stands to reason that they were a similarly important food item for the archaeological *U. maritimus* populations as they are today[38,39]. Therefore, we expect even substantial differences in *P. hispida* trophic levels among sites to have only a small effect on $\Delta^{15}N_{U.\ maritimus\ -\ P.\ hispida}$. Feeding at substantially different trophic levels is incompatible with modern *P. hispida* and *U. maritimus* population trophic levels and diet variability[38,39,54]. In addition, most other *U. maritimus* prey species feed on lower or similar trophic levels relative to *P. hispida*[8,80].

It is possible that due to the low intra-site sample size for both or either species, our site mean isotopic values do not capture the true means of the different populations. As the bones analysed are from individuals hunted or scavenged by humans, we cannot exclude differences in the segments of a *P. hispida* population hunted by humans and *U. maritimus*. For example, remains of *P. hispida* pups are very rare in archaeological assemblages[34,81]. *Ursus maritimus*, however, regularly preys on *P. hispida* pups and the contribution of pups to its diet may vary for different individuals, populations and with seal productivity[82,83]. As pups rely on their mother's milk, they effectively feed on a different trophic level leading to higher collagen $\delta^{15}N$ values in pups than adults[84]. Consequently, a higher consumption of *P. hispida* pups by *U. maritimus* relative to humans can lead to higher $\Delta^{15}N_{U.\ maritimus\ -\ P.\ hispida}$ values within an archaeological assemblage. Additional uncertainties for inter-site $\Delta^{15}N_{U.\ maritimus\ -\ P.\ hispida}$ values may arise from a higher contribution of migratory species such as *D. leucas*[85] to the diet of

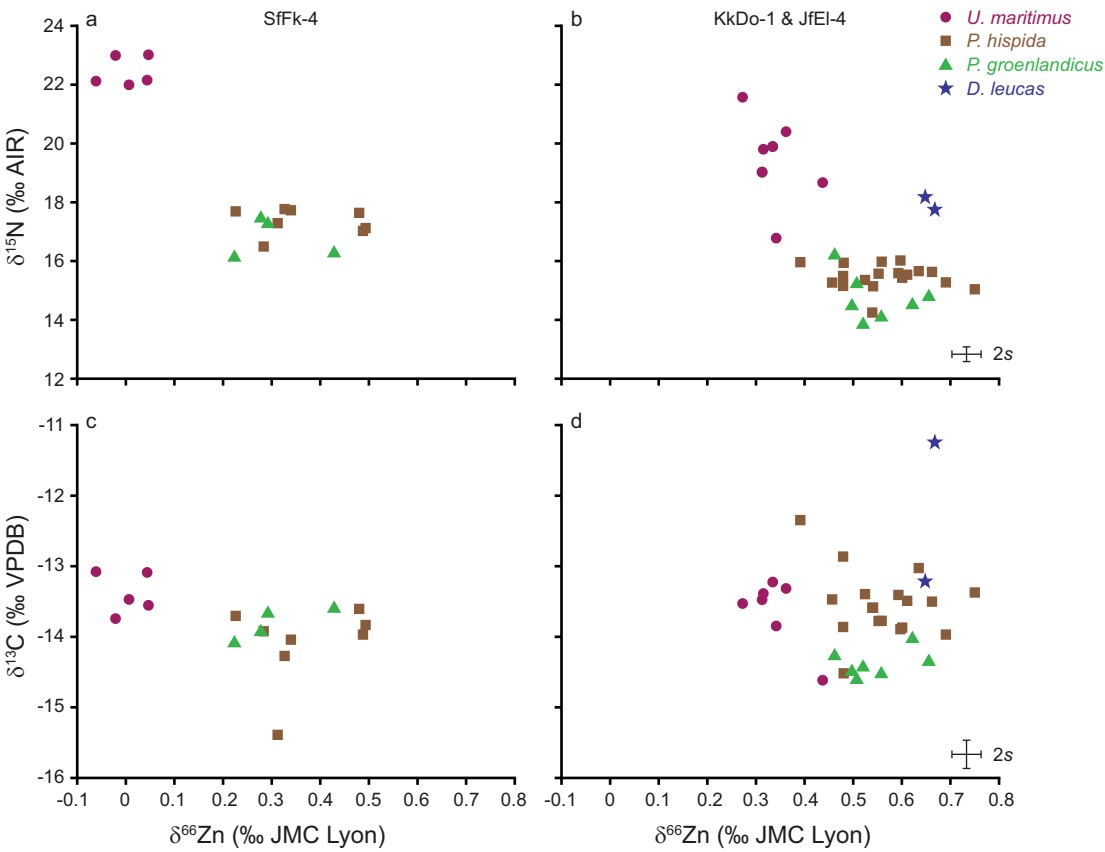

**Fig. 3 Isotope values for the SfFk-4 and KkDo-1 & JfEl-4 sites.** Isotopic composition ($\delta^{15}N$, $\delta^{13}C$ versus $\delta^{66}Zn$) of *U. maritimus* (magenta circles), *P. hispida* (brown squares), *P. groenlandicus* (green triangle) and *D. leucas* (blue stars) bones for the SfFk-4 (**a**, **c**) and combined KkDo-1 and JfEl-4 sites (**b**, **d**). For (**a**) and (**c**) we present $n = 5$ *U. maritimus*, $n = 8$ *P. hispida* and $n = 4$ *P. groenlandicus* bone samples and for (**b**) and (**d**) $n = 7$ *U. maritimus*, $n = 18$ *P. hispida*, $n = 7$ *P. groenlandicus* and $n = 2$ *D. leucas* bone samples. Error bars represent the measurement uncertainty.

certain *U. maritimus* populations[39]. Differences in the migratory behaviour of *P. hispida*[86] and/or *U. maritimus*[87] among sites could also lead to both species feeding along food chains with differing baseline $\delta^{15}N$ values and therefore variable $\Delta^{15}N_{U.\ maritimus\ -\ P.\ hispida}$ values within the assemblages. It remains, as of yet, unclear if and how physiological effects may influence $\delta^{66}Zn$ variability within a population. However, dietary differences as well as effects related to an archaeological assemblage (e.g., not capturing true population means) might have a similar effect on $\Delta^{66}Zn_{U.\ maritimus\ -\ P.\ hispida}$ as on $\Delta^{15}N_{U.\ maritimus\ -\ P.\ hispida}$.

For $\delta^{66}Zn$, two sites from the same geographic area close to the Labrador Sea (KkDo-1 and JfEl-4) have a markedly lower trophic spacing between *P. hispida* and *U. maritimus* $\delta^{66}Zn$ values of $-0.22$ and $-0.24‰$ (Fig. 2). Modern *U. maritimus* individuals from the area belong to the Davis Strait population[88]. In addition to *P. hispida* and contrary to most other *U. maritimus* populations, this one obtains a large percentage of its biomass from the consumption of harp seals (*Pagophilus groenlandicus*)[38,39]. However, bone $\delta^{66}Zn$ values of *P. hispida* and *P. groenlandicus* from the same site are indistinguishable (Fig. 3, Supplementary Fig. 9). Instead of the consumption of *P. groenlandicus*, the lower trophic discrimination factor for these sites may arise from unknown dietary contributions, population-specific physiological effects or unknown archaeological assemblage effects.

Unlike *P. hispida*, most *P. groenlandicus* leave their Canadian Arctic summering grounds, ahead of the formation of local pack ice in autumn[89,90]. However, when sympatric with *P. hispida*, *P. groenlandicus* feeds at a similar trophic level, consuming many of the same prey species, and both species show no statistical difference in muscle and liver $\delta^{15}N$ values[91]. Indeed, bones of both

seal species cover the same $\delta^{66}Zn$ range for the same location (Fig. 3). However, bones of *P. groenlandicus* from Hudson Strait (KkDo-1) have almost 1‰ lower mean $\delta^{15}N$ and $\delta^{13}C$ values than those of *P. hispida*, perhaps related to this species' seasonal southwards migration (Fig. 3b, d). In contrast, some *P. groenlandicus* individuals remain in west Greenland waters during winter[92,93], which may explain why bones of both seal species from eastern Ellesmere Island show a similar $\delta^{13}C$, $\delta^{15}N$ and $\delta^{66}Zn$ range (Fig. 3a, c). Due to its long turnover time, the bone collagen isotopic composition of *P. groenlandicus* likely represents an amalgamation of different food sources and local isotopic baseline values along their migration route and within their seasonal feeding grounds. *P. groenlandicus* bone $\delta^{66}Zn$ values do not seem to record migratory signals, again arguing for lower baseline variability or Zn isotope homogenisation within low trophic level organisms. Despite a very low samples size ($n = 2$) beluga whale (*Delphinapterus leucas*) $\delta^{66}Zn$ values fall within the same range as *P. hispida* and *P. groenlandicus* with slightly higher mean values (Fig. 3b, d). Indeed, all three species occupy a similar trophic level[8,91]. When sympatric with *P. hispida*, *D. leucas* typically has slightly lower soft tissue $\delta^{15}N$ values likely due to migrating between areas with differing baselines or a more offshore/pelagic foraging[8,76]. Here, *D. leucas* $\delta^{15}N$ values are higher than those of *P. hispida* and their $\delta^{13}C$ values are highly variable (Fig. 3). Instead of only reflecting this species' trophic level relative to *P. hispida*, their bulk collagen $\delta^{15}N$ and $\delta^{13}C$ values are likely influenced by the high mobility of this species[94] and its foraging in locations with different isotopic baselines.

The trophic levels of *U. maritimus*, *P. hispida* and *P. groenlandicus* are reflected by their bone collagen $\delta^{15}N$ and bone $\delta^{66}Zn$

values across the Arctic. The analysis of Zn isotopes, however, offers additional advantages for studying marine trophic ecology, not only due to its greater preservation potential in fossil material[16], but also due to the lower baseline controlled species-specific spatial isotopic gradients. The inclusion of $\delta^{66}$Zn analysis in ecological, archaeological and palaeontological studies may thus allow more robust interpretations of spatial and temporal trophic interactions. In addition, while both $\delta^{66}$Zn and $\delta^{15}$N generally record trophic levels, they do not record physiological and/or dietary effects equally, thus providing a strong incentive to combine $\delta^{66}$Zn with $\delta^{15}$N and $\delta^{13}$C analyses where possible.

In conclusion, this study compares archaeological bone $\delta^{66}$Zn values with traditional collagen $\delta^{15}$N and $\delta^{13}$C values for the same species across a large geographic area. Focussing on prey (*P. hispida*) and predator (*U. maritimus*) we investigate the baseline variability and trophic spacing of these dietary proxies. Our results show that:

(1) Overall, $\delta^{66}$Zn values shows less site-specific variability within a species, likely due to a lower baseline variability than for $\delta^{15}$N and $\delta^{13}$C. As such, $\delta^{66}$Zn values are particularly valuable for dietary studies on highly mobile species (or consumers thereof) and for comparing geographically and temporally distinct populations.

(2) We observe the expected trophic level spacing for collagen $\delta^{15}$N and bone $\delta^{66}$Zn values between *U. maritimus* and *P. hispida*. *U. maritimus* bone $\delta^{66}$Zn values are on average 0.32 ‰ lower than of its primary prey *P. hispida*.

(3) Bone $\delta^{66}$Zn values of the migratory species *P. groenlandicus* and *D. leucas* are consistent with respect to their known trophic levels when compared with *P. hispida* and *U. maritimus* values. In contrast, their collagen $\delta^{15}$N (and $\delta^{13}$C) values appear to document their relative trophic levels less precisely, likely influenced by variations in baseline isotopic compositions along their migration routes.

In ecological, archaeological and palaeontological research, trophic level estimations often rely exclusively on the $\delta^{15}$N tracer, sometimes biased by physiological, habitat and baseline effects. We demonstrate that the inclusion of $\delta^{66}$Zn analysis can provide otherwise inaccessible supplementary dietary information and more robust trophic level estimations.

## Methods

For this study, we compare $\delta^{66}$Zn, $\delta^{15}$N and $\delta^{13}$C values of 105 *P. hispida*, 47 *U. maritimus*, 11 *P. groenlandicus* and 2 *D. leucas* archaeological bone samples from across the Arctic (Supplementary Data 1). The data presented herein includes already-published $\delta^{66}$Zn values from an archaeological site (QjJx-1) on Little Cornwallis Island[15]. Additional $\delta^{66}$Zn values analysed for this study comprise 93 *P. hispida* bone samples from 13 archaeological sites (12 locations) and 37 *U. maritimus* bone samples from 11 archaeological sites (8 locations) as well as *P. groenlandicus* and *D. leucas* samples (2 sites, 1 site, respectively). For 6 of the 17 sites analysed here (RbJu-1, PaJs-13, QkHn-13, QjJx-1, KTZ sites), $\delta^{15}$N and $\delta^{13}$C values were already published elsewhere[4,15,30]. In addition, $\delta^{15}$N and $\delta^{13}$C values from one *P. hispida* sample from the NkRi-3 and OkRn-1 sites were already published elsewhere (Sample Nr. 4945, 9535)[31]. For the sites JfEl-4, KcFs-2, NkRi-3, seal bones analysed are identified as most likely *P. hispida*, but we cannot completely rule out that some samples may also come from other Phocidea (Supplementary Discussion 3.1). A single walrus (*Odobenus rosmarus*) bone and a potentially misidentified *D. leucas* bone from the JfEl-4 site were also measured and are compared to previously measured *O. rosmarus* bones from the QjJx-1 site[15] and our $\delta^{66}$Zn data from other species and sites in the Supplementary Discussion 3.4. Additional information and references regarding the archaeological context of the samples and sites are provided in the Supplementary Note 1.2 and Supplementary Table 3.

**Zinc analysis.** All samples' surfaces were mechanically abraded (cleaned) to avoid sediment contamination, using a dental drill equipped with a diamond-tipped burr. Approximately 10 to 50 mg chunks were then sampled using a diamond-tipped cutting wheel. The chunks were then ultrasonicated in ultrapure water (Milli-Q water) for 5 min and dried in a drying chamber for a few days at 50 °C. Bone

samples and reference materials NIST SRM 1400 and NIST SRM 1486 were subjected to different dissolution methods (HCl and $HNO_3$) to investigate the impact of the organic bone phases on its Zn isotope signal (Supplementary Methods 2.1, Supplementary Discussion 3.2). The column chromatography steps (3.1.2) for quantitative recovery of sample Zn[95,96] was the same for all samples regardless of the dissolution methods used. Each column chromatography batch ($n = 15$) included up to 13 samples, one chemistry blank and at least one reference standard (SRM 1400 and/or 1486).

Zn purification was performed in two steps, following the modified ion exchange method adapted from Moynier et al.[96], first described in Jaouen et al.[13]. Each step included AG-1 × 8 resin that was cleaned and conditioned prior to sample loading. One ml of AG-1 × 8 resin (200–400 mesh) was placed in 10 ml hydrophobic interaction columns (Macro-Prep® Methyl HIC). Resin cleaning involved 5 ml 3% $HNO_3$ followed by 5 ml ultrapure water. These cleaning steps were repeated. The resin was then conditioned with 3 ml 1.5 M HBr. After loading, 2 ml HBr were added for matrix residue elution followed by Zn elution with 5 ml $HNO_3$. Following the second column step, the solution was evaporated for 13 h at 100 °C and the residue re-dissolved in 1 ml 3% $HNO_3$.

Zn isotope ratios were measured using a Thermo Fisher Neptune MC-ICP-MS at the Max Planck Institute for Evolutionary Anthropology (Leipzig, Germany) and a Thermo Fisher Neptune Plus MC-ICP-MS at the Géosciences Environnement Toulouse - Observatoire Midi-Pyrénées (Toulouse, France). Instrumental mass fractionation was corrected by Cu doping following the protocol of Maréchal et al.[95] and Toutain et al.[97]. The in-house reference material Zn AA-MPI was used for standard bracketing. $\delta^{66}$Zn values are expressed relative to the JMC-Lyon reference material. Analysed sample solution Zn concentrations were close to 300 ppb as was the Zn concentration used for the standard mixture solution. Zn concentrations in the respective samples were estimated following a protocol adapted from one used for Sr by Copeland et al.[98], applying a regression equation based on the $^{64}$Zn signal intensity (V) of three solutions with known Zn concentrations (150, 300 and 600 ppb). $\delta^{66}$Zn uncertainties were estimated from standard replicate analyses and ranged between ±0.01‰ and ±0.03‰ (1 SD). Additional reference materials SRM 1486 and SRM 1400 were analysed alongside the samples. SRM reference materials and samples show a normal Zn mass dependent isotopic fractionation, i.e., the absence of isobaric interferences, as the $\delta^{66}$Zn vs. $\delta^{67}$Zn and $\delta^{66}$Zn vs. $\delta^{68}$Zn values fall onto lines with slopes close to the theoretic mass fractionation values of 1.5 and 2, respectively.

**Carbon and nitrogen isotope analysis.** Bone surfaces were cleaned with a dental drill equipped with a diamond-cutting wheel. Subsamples of bone chunks (100–200 mg) were demineralised in 0.5 M HCl at 4 °C. After demineralisation, samples were rinsed to neutrality with Type I water (resistivity > 18.2 MΩ cm). Any bone samples with dark colouration were treated with 0.1 M NaOH for successive 30 min treatments under sonication at room temperature until the solution no longer changed colour. The samples rinsed to neutrality with Type I water and then the insoluble collagen residue was solubilised in ~8 ml of 0.01 M HCl at 75 °C for 48 h. The resulting solution containing the solubilised collagen was filtered through a 5–8 µm filter and then filtered using a Microsep® 30 kDa molecular weight cut-off (MWCO) ultrafilter (Pall Corporation, Port Washington, NY) to remove low molecular weight compounds[99]. The >30 kDa fraction was freeze-dried, and the collagen yield was calculated.

Carbon and nitrogen isotopic and elemental compositions were determined using an IsoPrime continuous flow isotope-ratio mass spectrometer (CF-IRMS) coupled to a Vario Micro elemental analyser (Elementar, Hanau, Germany). Carbon and nitrogen isotopic compositions were calibrated relative to the VPDB and AIR scales, respectively, using a two-point calibration anchored by USGS40 (accepted $\delta^{13}$C − 26.39 ± 0.04‰, $\delta^{15}$N − 4.52 ± 0.06‰) and USGS41 (accepted $\delta^{13}$C + 37.63 ± 0.05‰, $\delta^{15}$N + 47.57 ± 0.11‰)[100]. Standard uncertainty was determined to be ±0.20‰ for $\delta^{13}$C and ±0.25‰ for $\delta^{15}$N[101]. Additional details are provided in the Supplementary Methods 2.2 and Supplementary Tables 4–6.

**Statistics and reproducibility.** All 144 samples analysed herein for $\delta^{66}$Zn and all 102 samples analysed for $\delta^{13}$C and $\delta^{15}$N were measured when possible, at least in duplicate with a mean standard deviation for sample replicates of ±0.01‰, ±0.12‰ and ±0.14‰, respectively.

Analysis of variance (ANOVA) were performed across the dataset in order to determine statistical differences in $\delta^{13}$C, $\delta^{15}$N and $\delta^{66}$Zn values between *P. hispida* populations. A single *P. hispida* specimen from Little Cornwallis Island was excluded from the statistical analysis ($\delta^{66}$Zn = 1.00‰, from Jaouen et al.[15]), as it could disproportionately influence the analysis (see Supplementary Fig. 4 versus 5). It was singled-out as an extreme outlier lying more than three times the interquartile range above the third quartile, both within-site and for the whole *P. hispida* dataset. Where variance was found to be significant, post hoc Tukey pair-wise comparisons were carried out to determine which populations were significantly different from each other in terms of their $\delta^{13}$C, $\delta^{15}$N and $\delta^{66}$Zn values. To adhere to ANOVA's assumptions, each *P. hispida* populations' $\delta^{13}$C, $\delta^{15}$N and $\delta^{66}$Zn datasets underwent visual inspection to check for normally distributed and homogeneous residuals, as well as tested for equal variance using Levene's test. Accordingly, we report the results of ANOVAs and post hoc Tukey pair-wise comparisons (Supplementary Figs. 4–7, Supplementary Data 3). As the

$\delta^{15}N$ dataset violated the equal variance assumption, an alternative Welch ANOVA was conducted instead, with post hoc Games-Howell pair-wise comparisons. In order to investigate the homogeneity of $\delta^{66}Zn$ values within the Arctic relative to $\delta^{15}N$ values, a series of Levene's test for equal variance (with Bonferroni correction) was performed on Zn and N isotope values between *P. hispida* and *U. maritimus*, as well as between sites for which data are available for both species. All statistical analyses were conducted using the free program R software[102].

**Reporting summary**. Further information on research design is available in the Nature Research Reporting Summary linked to this article.

## Data availability

All data generated during this study are included in this published article (and Supplementary Data 1–3). Provenance information including sample ID, Bordon code, sampling location, feature and sample source are given in Supplementary Data 1, with additional site information in Supplementary Table 3.

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

## Acknowledgements

The Max Planck Society funded the cost of Zn isotope analyses and salary. P.S. was supported by the SSHRC Insight Development grant (grant number 430 − 2014 − 00046). Salary support was provided to K.J., P.M. and N.B. by the ERC ARCHEIS project (Grant number 803676) and the DFG PALEODIET project (Grant number 378496604). We thank S. Steinbrenner and M. Trost (Department of Human Evolution, Max Planck Institute for Evolutionary Anthropology, Leipzig) for technical support.

## Author contributions

K.J. and J.M. designed the study. P.S., K.J. and J.M. selected the sample material. P.S., M.R. and C.H. performed the $\delta^{13}C$ and $\delta^{15}N$ analyses. J.M. and N.B. performed the $\delta^{66}Zn$ analyses at the MPI EVA Leipzig. J.M., K.J. and P.M. performed the $\delta^{66}Zn$ analyses at the CNRS in Toulouse. N.B. performed the statistical analysis. J.M., K.J., N.B., P.S., M.R. and J.-J.H., analysed and interpreted the data. J.M. wrote the initial manuscript. All authors contributed to editing the final version of the manuscript.

## Funding

## Competing interests

The authors declare no competing interests.
