## [Transparent Peer Review File · Communications Biology]

Reviewers' comments:

Reviewer #1 (Remarks to the Author):

1. Brief summary of the manuscript

The manuscript presents a research on a new method to estimate marine mammals' trophic level from bones collected from middens by analysing their $\delta^{66}\text{Zn}$ value. The research is original, well-conceived and well replicated, being also relevant to marine ecology and paleoecology fields. The authors have demonstrated that $\delta^{66}\text{Zn}$ values from bones can be used as a proxy of trophic level that is less affected by regional baseline values and is better preserved through time (compared to the widely used $\delta^{13}\text{C}$ and $\delta^{15}\text{N}$ values of collagen). The authors were able to provide equations for the estimates of trophic level of marine mammals based on bone $\delta^{66}\text{Zn}$ and collagen $\delta^{15}\text{N}$ values. They have also showed that bone $\delta^{66}\text{Zn}$ values are not affected by collagen-bound Zn, allowing easy analysis of both bone $\delta^{66}\text{Zn}$ and collagen $\delta^{13}\text{C}$ and $\delta^{15}\text{N}$ values. They have described some of the shortcomings of the work in the discussion and supplementary discussion, mainly related to species misidentification and the effects of migrations and diet diversity to isotopic values.

2. Overall impression of the work

It is really good to see research that expands the biochemical toolbox of marine ecologists, especially when they can be used to analyse samples from animals that lived hundreds or thousands of years ago. However, some aspects of isotopic and marine ecology should be better introduced and discussed (see specific comments). Studying past food webs can be challenging, researchers must make assumptions regarding environmental conditions and species behaviour living in habitats that no longer exist. The work clearly shows that Zn isotopes from bone can reliably show differences between predators and prey, with less variability related to different baselines supporting geographically distant food webs. This is a novel approach in marine ecology and paleoecology that can be used to answer a large array of important questions regarding temporal shifts in ecological baselines due to climatic and anthropogenic activities. I think that since this is the first archaeological study of its kind, perhaps more powerful biochemical tools should have been included, such as compound specific stable isotope analysis of amino acids (CSIA-AA) in collagen, so the authors would be able to better distinguish between trophic and environmental forcing in $\delta^{15}\text{N}$ and $\delta^{13}\text{C}$ variability.

Although the use of Zn isotope looks promising for ecological and palaeoecological studies, I think there are a couple of things that should be included when weighing this potential as a biogeochemical tool. For example, compound specific stable isotope analysis of amino acids (CSIA-AA) have been proven to be a powerful tool in both ecology and paleoecology fields, with the potential to describe isotope values of consumers and producers from a single sample 1–3. The present manuscript does not discuss or compare the use of these tools, and although running CSIA-AA can be expensive, the present paper also does not describe the costs of $\delta^{66}\text{Zn}$ analysis, an important information for such a novel approach in marine paleoecology. In my opinion, the use of $\delta^{66}\text{Zn}$ over other tools such as CSIA-AA, can only be justified (at the present), due to its lower costs or in habitats with high levels of collagen decomposition and diagenesis (I do not believe the Arctic can be characterized as such). As the title of the paper starts with "ADVANTAGES of zinc isotopes...", I think these two points should be better discussed in the paper, as well as the mention of CSIA-AA.

$\delta^{66}\text{Zn}$ values are distributed differently throughout the body of living animals, including between bones and other soft tissues, as shown by references number 19, 20 and 21 in the manuscript. Because of that, it is likely that animals feeding on selective body parts, e.g. the carnivorous *U. maritimus* (and even *P. hispida* in some cases) will present a different $\delta^{66}\text{Zn}$ trophic fractionation than species feeding on whole organisms, such as many fish, dolphins and sea birds. Therefore, it is likely that the findings of the present work, which were very interesting, relevant and well presented by the authors, can only be applied to study the ecology of specific carnivores such as *U. maritimus*. On these grounds, I would suggest the authors to address this issue in the broader context of marine ecosystems, or to change the title from "...proxy in marine ecology" to something more specific to the study of carnivores. Furthermore, even in the case of *P. hispida*, differences in diet and prey ingestion (either whole or soft tissue only) would affect their bone $\delta^{66}\text{Zn}$ value. *P. hispida* are likely to ingest

crustaceans, cephalopods, amphipods and even small fish whole, but in the case of larger fish prey, only soft body parts such as muscle and organs might be ingested, altering its bone $\delta^{66}\text{Zn}$ value. This is somewhat discussed in the supplementary discussion when the paper talks about the possible effect of different diets and geographical location, but in my opinion this should be discussed in more detail. A lot of the results, especially the ones from the "Species-specific isotopic variability" section, are actually given in the discussion. I suggest those to be added to the results section before being better explained in the discussion.

The supplementary material has a lot of overlapping information when comparing to the main text, especially the supplementary introduction. I would suggest for the authors to shorten this section and make the writing more straight forward, always remembering that the readers have already read the main text.

Throughout the text when the authors cite information from the supplementary material, especially parts of the supplementary discussion (which is extensive), the specific location of the information, like the section name or number, should be given. This way the reader interested in that specific information do not need to read the whole supplementary discussion to find it.

3. Specific comments, with recommendations for addressing each comment

Line 35: I believe that the references 3 and 4 did not investigate differences in faunal assemblages (i.e. community composition) between modern and fossil samples. In turn, Misarti et al. (2009) and Szpak et al. (2017) provided evidence of changes in isotopic values of collagen samples from marine organisms, possibly due to long-term environmental changes.

Line 43: The paper does not describe what the isotopes are used for the calculation of $\delta^{66}\text{Zn}$, apart for a sentence in the abstract. The reader might assume that the second most abundant isotope ($\delta^{64}\text{Zn}$) is used, but this should be explained when the authors first mention $\delta^{66}\text{Zn}$.

Additionally, while the fractionation of $\delta^{13}\text{C}$ and $\delta^{15}\text{N}$ and their use in marine ecology is straightforward, since there are only two stable isotopes to be measured, in the case of Zn there are five, four which have relatively high abundance in natural systems. As a reviewer and reader, I wonder how much ecological information is lost by not accounting for those isotopes. I suggest the authors to add some information in this respect, either in the introduction or discussion.

Line 47: Some authors suggest that the term "trophic position" be used as a combination of a species' trophic level and the resources supporting its diet, therefore resulting from information from both $\delta^{13}\text{C}$ and $\delta^{15}\text{N}$ isotopes. I see here that the paper uses "trophic position" and "trophic level" interchangeably. If this is the case please make it clear to the reader, otherwise use only one of the terms throughout the text.

Line 68: Please add citation to "Particularly for bone collagen, with its long turnover time"

Line 74: Maybe it is worth including here the information available in the supplementary material with references to studies indicating higher Zn concentrations is coastal than in oceanic zones.

Line 75: What do you mean by bulk marine Zn? Are you still talking about dissolved Zn or do you mean Zn from POM, or maybe both? Also at what depths? You have not mentioned any values for dissolved Zn at waters above 500m yet. I understand that this information is available at the review by Little et al. (2013), but this should be more easily available to the reader.

Line 79: Did you mean surface waters?

Line 92: This last section needs rephrasing, please better explain the advantage of analysing tissues from high trophic level species. Marine consumers, especially top predators are good integrators of multiple food web channels that are supported by different basal organic matter sources. That's why they can be representative of the basal sources supporting the whole food web and also show less variability in isotope values.

Line 104: I think a little bit more information about this experiment should have been given earlier, maybe together with the aims of the study. For me, the implications of this experiment are quite important to the field, since it allows scientists to easily analyse both collagen $\delta^{13}\text{C}$, $\delta^{15}\text{N}$ and bone $\delta^{66}\text{Zn}$ from a single sample. This is an advantage of this new methods, as the title says, so I think it should be better introduced to the reader.

Line 127: What is the typical trophic level variability? Were they calculated with Supplementary equation 1 and presented in Supplementary table 7? This is not very clear to the reader. Furthermore, in line 82 the paper states that: "we aim at exploring Zn isotopes as a reliable tracer of marine trophic levels", but there is no direct comparison between Zn isotopes and trophic levels of the species analysed in the main text. Instead this information is only given in the form of supplementary tables and in the supplementary figure 5. I understand that, in the way trophic level was calculated in the present manuscript (not taking into account regional and temporal differences in isotopic baseline, tissue turnover rates and isotopic fractionations), $\delta^{15}\text{N}$ values behave exactly like estimated trophic level. Although this justifies the chosen approach, this should be made clear to the reader that are not familiar with the dynamics of isotopic ecology.

Line 144: Although I understand that nitrogen isotope can be a proxy of trophic level, trophic position (or trophic level) is not measured in ‰, like this sentence states (but see comment above).

Line 157: If the differences in $\delta^{66}\text{Zn}$ values observed in the QjJx-1 and QkHn-13 sites are related to the high environmental variability within the CAA, how the non-variability in the rest of the Archipelago, i.e. in the PaJs-13 and RbJu-1 sites, are explained?

Line 174: The results from the statistical tests have already been shown in the results section, I would suggest not to display them again in the discussion.

Line 164: Do the authors think that a better understanding of $\delta^{66}\text{Zn}$ baselines throughout the ocean (or $\delta^{66}\text{Zn}$ isoscape) is needed for the proper use of $\delta^{66}\text{Zn}$ in marine ecology?

Line 186: I think this first sentence is a little problematic and need a bit of attention. The fact that there is a lot of variability between a consumer's and a predator's $\delta^{15}\text{N}$ values does not reflect the challenges of estimating trophic level more than the diet variability of both species. *U. maritimus* and *P. hispida* have broad diets, as discussed in the supplementary information. Furthermore, these species can move long distance, consuming prey from different regions. For example, *U. maritimus* can move 12km a day (Ferguson et al 2001 "Activity and Movement Patterns of Polar Bears Inhabiting Consolidated versus Active Pack Ice" Arctic Institute of North America), while *P. hispida* can cover an area of 5844km² in around 256 days (Harwood et al 2015 "Seasonal Movements and Diving of Ringed Seals, *Pusa hispida*, in the Western Canadian Arctic, 1999-2001 and 2010-11" Arctic Institute of North America). Because of the diet variability of every species in the food web, it is only possible to estimate trophic level by knowing the diet composition of every individual, or by using isotopic values of primary producers and assuming equal trophic fractionation throughout the whole food web. In my opinion, the difference of $\delta^{15}\text{N}$ values between *P. hispida* and *U. maritimus* might only show how those two species are connected or not to the same food web and supported by similar basal resources. As mentioned before, although this is out of the scope of this manuscript, these uncertainties, described briefly in the end of this paragraph and in the next, could have been disentangled by applying other biochemical tools such as CSIA-AA. Even with diet and baseline uncertainties, if the authors wish to display how the fractionation of $\delta^{15}\text{N}$ and $\delta^{66}\text{Zn}$ between *P. hispida* (or other prey species) and *U. maritimus* are related, the regression between predator-prey $\delta^{15}\text{N}$ and predator-prey $\delta^{66}\text{Zn}$ values could be added to the manuscript.

Line 253: Maybe I am missing something here, but what kind of bone was used in this research and how were them identified at species level?

Line 332: Did all the compared groups show a normal distribution and equal variances? In lines 122, 148 and 173 the paper states that variances were unequal between sites for $\delta^{15}\text{N}$ values of *P. hispida* and *U. maritimus*. What was the approach when the assumptions could not be met?

Figure 1: Please include latitude and longitude graticulates, scale bar and a North arrow to better indicate the sampling locations to the reader. This comment concern all the maps in the manuscript.

Table 1: Since there are only two samples of *D. leucas*, that correspond to the maximum and minimum isotope values, in my opinion the calculation of SD for that species is redundant.

Supplementary Material:

Line 160: I might have missed something, but at this point I have already gone through the whole manuscript but I still do not quite grasp the importance of the experiment described in the supplementary figure 1. The authors could describe this aspect of the work better in the introduction or in the supplementary introduction

Line 179: This information should be part of the introduction, see comment above

Line 189: I understand this might be out of scope for this manuscript, but what about other techniques that could also be applied to the collagen? Could CSIA-AA be also used in small samples?

Line 194: Is this sentence related to the samples analysed in the present study? If so please specify

Line 216: This goes back to my comment about how those bones were identified and what specific bones were used in the analysis.

Line 270: Temporal variability in isotopic values of primary producers is also an issue that can be discussed here, since samples collected hundred and thousand years apart were analysed in the present study. Would be interesting to know how oceanic $\delta^{66}\text{Zn}$ baselines are expected to vary throughout this time frame and this can be discussed.

Line 280: This sentence needs rephrasing, as this is only true if all prey items are supported by the same basal organic matter source.

Line 283: Food webs have changed quite a lot in the past 1000 years, especially due to human interactions. Different ranges in isotopic values could be a results of those changes, producing more diverse, less overlapping food webs. See Saporiti et al 2014 "Longer and Less Overlapping Food Webs in Anthropogenically Disturbed Marine Ecosystems: Confirmations from the Past" PLoS One. Furthermore, ice cover can influence diet and ecology of both predator and prey species (Wing et al 2020 "Penguins and Seals Transport Limiting Nutrients Between Offshore Pelagic and Coastal Regions of Antarctica Under Changing Sea Ice" Ecosystems), which is known to have changed in the past millennia. Is this sentence still relevant in this context?

Line 329: I understand the advantages of $\delta^{66}\text{Zn}$ data, but as one of my earlier comments, nitrogen isotopic differences between those two species cannot be used as a trophic level estimates. Difficulties arise mainly from lack of baseline data from primary producers, diet, metabolism and consumption rate (prey availability) variability, for both consumer and predator.

Line 355: The information in this section should have been provided much earlier in the manuscript, in order to provide the basis for the trophic level estimates.

Line 374: Are those results in line with the published literature?

References

1. Whiteman, J. P., Smith, E. A. E., Besser, A. C. & Newsome, S. D. A guide to using compound-specific stable isotope analysis to study the fates of molecules in organisms and ecosystems. *Diversity* 11, 1–18 (2019).
2. Sabadel, A., Durante, L. & Wing, S. Stable isotopes of amino acids from reef fishes uncover Suess and nitrogen enrichment effects on local ecosystems. *Mar. Ecol. Prog. Ser.* 647, 149–160 (2020).
3. Evershed, R. P. et al. Compound-Specific Stable Isotope Analysis in Ecology and Paleoecology. in *Stable Isotopes in Ecology and Environmental Science* 480–540 (Blackwell Publishing Ltd, 2008). doi:10.1002/9780470691854.ch14

I congratulate the authors for the manuscript, which was a pleasure to read.

Leonardo Maia Durante

Reviewer #2 (Remarks to the Author):

The authors present $\delta^{66}\text{Zn}$, $\delta^{15}\text{N}$ and $\delta^{13}\text{C}$ values for four species of marine mammals from Arctic archaeological sites in order to assess how well the isotopic proxies perform against one another as trophic indicators. This builds on the work of Jaouen et al. 2016, expanding that analysis across many archaeological sites in the Arctic. The authors find that Zn is homogenous across for species across. A wide geographic range, whereas $\delta^{15}\text{N}$ is more variable, likely due to baseline effects. They suggest that $\delta^{66}\text{Zn}$ is a more reliable trophic proxy, but part of this analysis hinges on $\delta^{15}\text{N}$ derived trophic levels, which seems contradictory and needs further explanation.

I think this is a great study, but I think it should be published in a longer format journal. All of this work with nontraditional isotopes is exciting and welcome contribution to the literature, but the authors need to clarify the discussion of their results, especially how $\delta^{15}\text{N}$ and $\delta^{66}\text{Zn}$ can be integrated when they claim that only one of those proxies is reliable. Also, a significant portion of the discussion and analysis have been bumped to the supplemental information, and I think this paper would be vastly improved by moving those points to the main text, especially the isotopic context section, which would have clarified many of the assumptions made in the main text. I also think the bone dissolution tests the authors did were incredibly useful, but it's hidden away in the supplement, as are the refined discussions on baseline isotopic values, preservation and diagenesis, non-dietary effects on trophic discrimination, etc.

Figure comments:

The extended data figures are impossible to read. The Y axis has been compressed to the point that you can't accurately read values from it, and the stacks on stacks of lines showing significance among the data as the outcomes of the ANOVA are extraordinarily challenging to read with $\delta^{15}\text{N}$ and $\delta^{13}\text{C}$ data. All of these extended data figures should be made into tables.

Results section

Four species of marine mammals are analyzed for C, N and Zn, but only the results of *P. hispida* are discussed in detail – why is that? Without thorough discussion of at least the other marine carnivore that was the focus of this study (*U. maritimus*), this is an incomplete section. It also seems that all

figures for this section are only presented in the Supplement and not the main body of the text – there should be a citation for Figure 1 here as well.

Discussion section

The authors lean heavily on the assumption that the base of Arctic food-web Zn must be more homogenous than for d13C and d15N in order to explain the lack of variation in Zn isotopes. This lack of variation is interesting, and definitely possible, but without direct measurements of d66Zn from the same plankton in this foodweb study, other possibilities must be discussed. It's possible (and also quite likely) that these higher carnivores have shifted prey sources and trophic positions through time, especially between the archaeological and modern samples. The authors should discuss this possibility in the text. They do discuss the varied diets of *P. hispida* and *U. maritimus* in the supplemental text. This discussion comes up later in the section (lines 186 – 199), but I think it should appear sooner for clarity.

The authors state that d66Zn is more reliable than d15N, but then use d15N to determine trophic level for these species in order to determine trophic level discrimination factors for Zn across the samples. This seems contradictory, and comes across as the authors using d15N only to validate the other isotope system, while writing off the d15N data as unreliable. This needs to be clarified and/or rewritten.

I think something that would clarify this paper and make it easier to follow would be a summary table of trophic position estimates for all species in the study from both isotopic investigations (from this and previous studies) and also from the ecological literature. It is extremely challenging to follow the discussion because it's never very clear where inferred trophic position for each species is coming from, and whether or not that differs from previous research.

Line 176: (and diet?) is a more casual writing expression, I would rewrite this to something more formal, eg, "...values for a specific taxon, and possibly diet,...".

Reviewer #3 (Remarks to the Author):

Review of "Advantages of zinc isotopes as a new dietary proxy in marine ecology" by McCormack et al.

This manuscript describes the utility of using 66Zn isotopic compositions in archeological samples to determine relative trophic positions in the marine food web, in this case for polar bears and their main prey, seals. The main premise is that d66Zn values may be as useful or, in some cases, more useful than using d15N values as is traditionally done for ecological studies. This manuscript is of high quality, is well written and concise, and I don't see any flaws in the methodology, although I admit I am not an expert in non-traditional stable isotopes. I very much enjoyed reading it. It certainly may be a very useful technique for certain types of samples. I recommend it for publication either as is or subject to minor revision. Consequently, I only have a few comments for the authors as indicated below.

To me, the advantage of 66Zn is use in archeological samples where little is known about baseline or geographic variation in d15N values, or when collagen is absent. For "fresh" samples, d15N values should give the same information, but are much cheaper and far less labour intensive than the methods described for measurement of d66Zn values. This is not meant as a criticism, but rather I think the authors may want to emphasize that this method is ideally suited for archeological studies. As it is, the tone of the manuscript implies that this new method may be as useful as traditional stable isotope analyses for fresh samples. Calculated trophic levels and their relative error using d66Zn and d15N values are basically identical and indistinguishable (Supplementary Tables 7 and 8) so I think that for modern samples d15N analyses would be preferred.

The archaeological sites examined vary in age from 325-4100 y, with most samples spanning an age of 200 y or so. What effects would the authors expect from this? For instance, is it possible that the time span between individual samples (i.e. two samples from the same site could be up to 200 years apart in age) collectively serves to smooth out the differences between $\delta^{66}\text{Zn}$ values more so than $\delta^{15}\text{N}$ values of collagen? I'm fairly certain that the variation in baseline $\delta^{15}\text{N}$ or $\delta^{13}\text{C}$ values in the arctic over a period of 100's years is unknown.

Title. Again, It is unclear from the title that this study deals with archeological samples and I suggest that this be included. At first glance it may be incorrectly assumed that this study deals with current (i.e. fresh) ecological samples.

Introduction. I. 63. Baseline isotopic variations can also be compensated by compound specific amino acid isotopic analyses - see Webb et al, 2015 for an example application of this technique to collagen.

Extended Data Figures. I'm not sure there is any way to graphically express the various p-values for the statistical tests, but these figures are certainly confusing at best. I'm uncertain of the structure of the published article, but I would suggest that these be omitted from the main article and included in the supplementary material.

Supplementary Information.

Baseline carbon and nitrogen isotope variability recorded in *P. Hispida* bones. I suggest the authors further elaborate on possible reasons behind this variability. For example, terrestrial carbon influx from rivers, such as the McKenzie, may serve to drive down marine $\delta^{13}\text{C}$ values.

Figure 1. Demineralized collagen intact. Should be intact?

Author response

We acknowledge the very thorough and constructive feedback given by all three reviewers. Implementing the reviewers' comments significantly strengthened our arguments.

In the following we discuss the comments made by each referee (original remarks are in black, our rebuttal/comments in blue). We also numbered the reviewer comments (green numbers), allowing us to refer to comments of other reviewers and our answers. When referring to changes made with line numbers, we refer to the tracked changes versions of the manuscript and Supplementary Information unless otherwise specified.

General comment:

Reviewer number 2, but also reviewer 3 (comment 3.2), seem to be under the impression, that we intend to introduce $\delta^{66}\text{Zn}$ here as a “better” trophic level proxy or replacement for $\delta^{15}\text{N}$. This comes as a surprise to us, as we highlight in the abstract (original lines 25-26), the introduction (original lines 42-45), the discussion (original lines 224-228) and the conclusion (original lines 248-251) that we see this method as a much-needed addition to $\delta^{15}\text{N}$ analyses, as both reflect trophic levels, likely equally well, but are independently controlled. Therefore, combining the methods can help verify the results of the other proxy and as each method has its advantages lead to more robust palaeoecological interpretations. The notion of only using one proxy instead of the other is perplexing to us, especially as multi-proxy approaches will always lead to a better understanding of a system in its entirety than when using only a single-proxy approach. We highlight this even more now in the introduction (lines 54-56, 122-125) and by adding more information about the dissolution experiment (which should enable combining both methods) to the introduction and results as suggested by reviewer number 1 and 2. We also deleted a line from the abstract which may have falsely given the impression that $\delta^{15}\text{N}$ may be unreliable.

Reviewer #1 (Remarks to the Author):

1. Brief summary of the manuscript.

1.1 The manuscript presents an research on a new method to estimate marine mammals' trophic level from bones collected from middens by analysing their $\delta^{66}\text{Zn}$ value. The research is original, well-conceived and well replicated, being also relevant to marine ecology and paleoecology fields. The authors have demonstrated that $\delta^{66}\text{Zn}$ values from bones can be used as a proxy of trophic level that is less affected by regional baseline values and is better preserved through time (compared to the widely used $\delta^{13}\text{C}$ and $\delta^{15}\text{N}$ values of collagen). The authors were able to provide equations for the estimates of trophic level of marine mammals based on bone $\delta^{66}\text{Zn}$ and collagen $\delta^{15}\text{N}$ values. They have also showed that bone $\delta^{66}\text{Zn}$ values are not affected by collagen-bound Zn, allowing easy analysis of both bone $\delta^{66}\text{Zn}$ and collagen $\delta^{13}\text{C}$ and $\delta^{15}\text{N}$ values. They have described some of the shortcoming of the work in the discussion and supplementary discussion, mainly related to species misidentification and the effects of migrations and diet diversity to isotopic values.

We thank the reviewer for the approval of our work.

2. Overall impression of the work.

1.2 It is really good to see research that expands the biochemical toolbox of marine ecologists, especially when they can be used to analyse samples from animals that lived hundreds or thousands of years ago. However, some aspects of isotopic and marine ecology should be better introduced and discussed (see specific comments). Studying past food webs can be challenging, researchers must make assumptions regarding environmental conditions and species behaviour living in habitats that no longer exist. The work clearly shows that Zn isotopes from bone can reliably show differences between predators and prey, with less variability related to different baselines supporting geographically distant food webs. This is a novel approach in marine ecology and paleoecology that can be used to answer a large array of important questions regarding temporal shifts in ecological baselines due to climatic and anthropogenic activities. I think that since this is the first archaeological study of its kind, perhaps more powerful biochemical tools should have been included, such as compound specific stable isotope analysis of

amino acids (CSIA-AA) in collagen, so the authors would be able to better distinguish between trophic and environmental forcing in $\delta^{15}\text{N}$ and $\delta^{13}\text{C}$ variability. Although the use of Zn isotope looks promising for ecological and palaeoecological studies, I think there are a couple of things that should be included when weighing this potential as a biogeochemical tool. For example, compound specific stable isotope analysis of amino acids (CSIA-AA) have been proven to be a powerful tool in both ecology and paleoecology fields, with the potential to describe isotope values of consumers and producers from a single sample 1–3. The present manuscript does not discuss or compare the use of these tools, and although running CSIA-AA can be expensive, the present paper also does not describe the costs of $\delta^{66}\text{Zn}$ analysis, an important information for such a novel approach in marine paleoecology. In my opinion, the use of $\delta^{66}\text{Zn}$ over other tools such as CSIA-AA, can only be justified (at the present), due to its lower costs or in habitats with high levels of collagen decomposition and diagenesis (I do not believe the Arctic can be characterized as such). As the title of the paper starts with “ADVANTAGES of zinc isotopes...”, I think these two points should be better discussed in the paper, as well as the mention of CSIA-AA.

We thank the reviewer for his compliments and for bringing up CSIA-AA as a valuable method to compensate environmental forcing on $\delta^{15}\text{N}$ and $\delta^{13}\text{C}$ values. As the reviewer pointed it out, CSIA-AA are a very expensive method, as well as quite time consuming and difficult to set up. In comparison, Zn purification only requires a 3 hours column step, repeated twice for 12 to 24 samples. Analyses on the mass spectrometer are also quite quick, 48 samples can be analysed within 24 hours. The analyses of 48 samples approximately cost 1000 euros, when we add up the cost of running the mass spectrometers, the resin, MilliQ water and reagents (ultrapure acids) used to prepare them. In comparison, the CSIA-AA analyses would cost us about 15 to 20 times more.

We had a first glance on how Zn isotopes could be interesting for marine ecology in our previous paper of Jaouen et al (2016) published in PloS One. We were therefore interested to go further in exploring its potential and we wanted to conduct a large-scale investigation, that we performed here with a corpus of 167 samples. We therefore did not consider running CSIA-AA here, as it was for us an entirely different topic, but the reviewer has a point that it could be extremely interesting to compare the two tracers, especially with $\delta^{15}\text{N}$ of AA, as it could help us identify what variations are actually due to diet or environmental factors. We will definitely consider it for a future study when we will have the time and the budget, possibly performing the analyses commercially with the University of Davis, or if we find interested collaborators. We included some information with references to CSIA-AA in the Introduction (lines 42-45). We already address the point that $\delta^{66}\text{Zn}$ is particularly useful in absence of collagen preservation in the Introduction (original manuscript line 42, with a citation). This manuscript, however, deals with the advantages of combining $\delta^{66}\text{Zn}$ with $\delta^{15}\text{N}$ and $\delta^{13}\text{C}$ values which allows more robust palaeoecological interpretations as demonstrated herein than by using “just” $\delta^{15}\text{N}$ and $\delta^{13}\text{C}$ values. We also highlight this now more in the introduction (lines 54-56, 122-125).

1.3 $\delta^{66}\text{Zn}$ values are distributed differently throughout the body of living animals, including between bones and other soft tissues, as shown by references number 19, 20 and 21 in the manuscript. Because of that, it is likely that animals feeding on selective body parts, e.g. the carnivorous *U. maritimus* (and even *P. hispida* in some cases) will present a different $\delta^{66}\text{Zn}$ trophic fractionation than species feeding on whole organisms, such as many fish, dolphins and sea birds. Therefore, it is likely that the findings of the present work, which were very interesting, relevant and well presented by the authors, can only be applied to study the ecology of specific carnivores such as *U. maritimus*. On these grounds, I would suggest the authors to address this issue in the broader context of marine ecosystems, or to change the title from “...proxy in marine ecology” to something more specific to the study of carnivores. Furthermore, even in the case of *P. hispida*, differences in diet and prey ingestion (either whole or soft tissue only) would affect their bone $\delta^{66}\text{Zn}$ value. *P. hispida* are likely to ingest crustaceans, cephalopods, amphipods and even small fish whole, but in the case of larger fish prey, only soft body parts such as muscle and organs might be ingested, altering its bone $\delta^{66}\text{Zn}$ value. This is somewhat discussed in the supplementary discussion when the paper talks about the possible effect of different diets and geographical location, but in my opinion this should be discussed in more detail.

This is true and an important issue. We address this issue now in the manuscript (lines 287, 296-303). Naturally, we were unable to analyse the various prey items of ringed seals. Based on the homogeneity of their $\delta^{66}\text{Zn}$ values independent of the site, we suspect, that they too have a relatively constant $\delta^{66}\text{Zn}$

trophic level discrimination factor relative to tissues of their prey. Whether or not this factor varies would only be possible to examine when individuals were control fed with specific food items (i.e., fish, cephalopods, amphipods, and so on). We anticipate these kinds of studies to emerge in the years to come. But for now, we can only point out the potential differences in trophic level fractionation factors. We do that now.

1.4 A lot of the results, especially the ones from the “Species-specific isotopic variability” section, are actually given in the discussion. I suggest those to be added to the results section before being better explained in the discussion.

We have added some of the information provided in the Species-specific isotopic variability section to the results (line 197-199). We do not display the results of the statistical tests in the results section anymore (see also response to comment 1.19).

1.5 The supplementary material has a lot of overlapping information when comparing to the main text, especially the supplementary introduction. I would suggest for the authors to shorten this section and make the writing more straight forward, always remembering that the readers have already read the main text. Throughout the text when the authors cite information from the supplementary material, especially parts of the supplementary discussion (which is extensive), the specific location of the information, like the section name or number, should be given. This way the reader interested in that specific information do not need to read the whole supplementary discussion to find it.

We have rephrased parts of the supplementary introduction and include the isotopic context section now in the main text, thereby avoiding repetition. We now also use section names when citing supplementary information to make it easier for the reader to find the information.

3. Specific comments, with recommendations for addressing each comment

1.6 Line 35: I believe that the references 3 and 4 did not investigate differences in faunal assemblages (i.e. community composition) between modern and fossil samples. In turn, Misarti et al. (2009) and Szpak et al. (2017) provided evidence of changes in isotopic values of collagen samples from marine organisms, possibly due to long-term environmental changes.

We rephrased the sentence accordingly.

1.7 Line 43: The paper does not describe what the isotopes are used for the calculation of $\delta^{66}\text{Zn}$, apart for a sentence in the abstract. The reader might assume that the second most abundant isotope ($\delta^{64}\text{Zn}$) is used, but this should be explained when the authors first mention $\delta^{66}\text{Zn}$.

We added that we measure the $^{66}\text{Zn}/^{64}\text{Zn}$ ratio again in the introduction. Still, this comment comes as a surprise, as we do mention (as stated by the reviewer) which isotopes are being used to calculate $\delta^{66}\text{Zn}$ in the abstract, when first used. In any case, by definition, the delta value is given as the ratio of the abundance of the heavy to light isotope in a sample relative to a reference standard. Zinc only has one stable isotope lighter than ^{66}Zn (i.e., ^{64}Zn).

1.8 Additionally, while the fractionation of $\delta^{13}\text{C}$ and $\delta^{15}\text{N}$ and their use in marine ecology is straightforward, since there are only two stable isotopes to be measured, in the case of Zn there are five, four which have relatively high abundance in natural systems. As a reviewer and reader, I wonder how much ecological information is lost by not accounting for those isotopes. I suggest the authors to add some information in this respect, either in the introduction or discussion.

No ecological information is lost by reporting only $\delta^{66}\text{Zn}$. The basis of stable isotope research is mass-dependent fractionation. Mass-dependent fractionation affects all stable Zn isotopes, leading to a linear relationship between $\delta^{66}\text{Zn}$ vs. $\delta^{67}\text{Zn}$ and $\delta^{66}\text{Zn}$ vs. $\delta^{68}\text{Zn}$. We report this mass dependency (documented as $\delta^{66}\text{Zn}$ vs. $\delta^{67}\text{Zn}$ and $\delta^{66}\text{Zn}$ vs. $\delta^{68}\text{Zn}$) as an argument against significant isobaric interferences affecting our measurements (see original manuscript lines 301-303). Since ^{64}Zn and ^{66}Zn are the most abundant, it is common practice to report the $\delta^{66}\text{Zn}$ values as they have the best analytical precision.

1.9 Line 47: Some authors suggest that the term “trophic position” be used as a combination of a species’ trophic level and the resources supporting its diet, therefore resulting from information from both $\delta^{13}\text{C}$ and $\delta^{15}\text{N}$ isotopes. I see here that the paper uses “trophic position” and “trophic level” interchangeably. If this is the case please make it clear to the reader, otherwise use only one of the terms throughout the text.

We changed trophic position to trophic level throughout the manuscript and supplementary material to avoid any confusion.

1.10 Line 68: Please add citation to “Particularly for bone collagen, with its long turnover time”

Done.

1.11 Line 74: Maybe it is worth including here the information available in the supplementary material with references to studies indicating higher Zn concentrations is coastal than in oceanic zones.

We have rephrased parts of this section, now also including the isotopic context section in the main text.

1.12 Line 75: What do you mean by bulk marine Zn? Are you still talking about dissolved Zn or do you mean Zn from POM, or maybe both? Also at what depths? You have not mentioned any values for dissolved Zn at waters above 500m yet. I understand that this information is available at the review by Little et al. (2013), but this should be more easily available to the reader.

We have rephrased it to indicate that we are talking about bulk dissolved Zn. We also included the values for dissolved Zn in waters above 500 m. Please also see our comment to 1.5 and 1.11.

1.13 Line 79: Did you mean surface waters?

Yes. We changed it accordingly.

1.14 Line 92: This last section needs rephrasing, please better explain the advantage of analysing tissues from high trophic level species. Marine consumers, especially top predators are good integrators of multiple food web channels that are supported by different basal organic matter sources. That’s why they can be representative of the basal sources supporting the whole food web and also show less variability in isotope values.

Done. We rephrased accordingly.

1.15 Line 104: I think a little bit more information about this experiment should have been given earlier, maybe together with the aims of the study. For me, the implications of this experiment are quite important to the field, since it allows scientists to easily analyse both collagen $\delta^{13}\text{C}$, $\delta^{15}\text{N}$ and bone $\delta^{66}\text{Zn}$ from a single sample. This is an advantage of this new methods, as the title says, so I think it should be better introduced to the reader.

We have added additional information about the bone dissolution experiments to the introduction and expanded upon it in the results.

1.16 Line 127: What is the typical trophic level variability? Were they calculated with Supplementary equation 1 and presented in Supplementary table 7? This is not very clear to the reader. Furthermore, in line 82 the paper states that: “we aim at exploring Zn isotopes as a reliable tracer of marine trophic levels”, but there is no direct comparison between Zn isotopes and trophic levels of the species analysed in the main text. Instead this information is only given in the form of supplementary tables and in the supplementary figure 5. I understand that, in the way trophic level was calculated in the present manuscript (not taking into account regional and temporal differences in isotopic baseline, tissue turnover rates and isotopic fractionations), $\delta^{15}\text{N}$ values behave exactly like estimated trophic level.

Although this justifies the chosen approach, this should be made clear to the reader that are not familiar with the dynamics of isotopic ecology.

We are referring to the typical trophic level variability in $\delta^{15}\text{N}$ between predator and prey (e.g., +3.4 to +3.8 ‰) as shown in the literature^{8,16,17}. We rephrased this to clarify. We have now also rephrased parts of the introduction (lines 111-116), the discussion (lines 284-303) and supplementary discussion (lines 450-455) when dealing with the information provided by supplementary tables 7 and 8 and in the supplementary figure 5. See also our response to comments 2.1 and 2.6.

1.17 Line 144: Although I understand that nitrogen isotope can be a proxy of trophic level, trophic position (or trophic level) is not measured in ‰, like this sentence states (but see comment above).

This was not our intension. We simply state that Zn isotope variability between site mean values is lower than the observed mean variability between polar bear and ringed seal $\delta^{66}\text{Zn}$ values. We rephrased the sentence to avoid a similar misunderstanding again.

1.18 Line 157: If the differences in $\delta^{66}\text{Zn}$ values observed in the QjJx-1 and QkHn-13 sites are related to the high environmental variability within the CAA, how the non-variability in the rest of the Archipelago, i.e. in the PaJs-13 and RbJu-1 sites, are explained?

We added a sentence here, offering an alternative for the statistical variability for these sites (as also suggested by reviewer 2 comment 2.5). In adding this sentence here and rephrasing within this paragraph, we made clearer, that we cannot fully explain the variability between these sites, or the lack thereof for other sites. Due to the novelty of this method, no baseline $\delta^{66}\text{Zn}$ values exist yet, so we kept this section short, to not have a too speculative discussion.

1.19 Line 174: The results from the statistical tests have already been shown in the results section, I would suggest not to display them again in the discussion.

Thank you for pointing this out. We changed it accordingly.

1.20 Line 164: Do the authors think that a better understanding of $\delta^{66}\text{Zn}$ baselines throughout the ocean (or $\delta^{66}\text{Zn}$ isoscape) is needed for the proper use of $\delta^{66}\text{Zn}$ in marine ecology?

Please also see our response to comment 1.18. Our results indicate that baseline effects are probably less pronounced for $\delta^{66}\text{Zn}$ than $\delta^{15}\text{N}$ and $\delta^{13}\text{C}$. We do think that this should be investigated further. Naturally, this is beyond the scope of this project. Yet even without baseline information, we demonstrate that $\delta^{66}\text{Zn}$ can be properly used in marine ecology (see trophic discrimination between polar bears and their prey).

1.21 Line 186: I think this first sentence is a little problematic and need a bit of attention. The fact that there is a lot of variability between a consumer's and a predator's $\delta^{15}\text{N}$ values does not reflect the challenges of estimating trophic level more than the diet variability of both species. *U. maritimus* and *P. hispida* have broad diets, as discussed in the supplementary information. Furthermore, these species can move long distance, consuming prey from different regions. For example, *U. maritimus* can move 12km a day (Ferguson et al 2001 "Activity and Movement Patterns of Polar Bears Inhabiting Consolidated versus Active Pack Ice" Arctic Institute of North America), while *P. hispida* can cover an area of 5844km² in around 256 days (Harwood et al 2015 "Seasonal Movements and Diving of Ringed Seals, *Pusa hispida*, in the Western Canadian Arctic, 1999-2001 and 2010-11" Arctic Institute of North America). Because of the diet variability of every species in the food web, it is only possible to estimate trophic level by knowing the diet composition of every individual, or by using isotopic values of primary producers and assuming equal trophic fractionation throughout the whole food web. In my opinion, the difference of $\delta^{15}\text{N}$ values between *P. hispida* and *U. maritimus* might only show how those two species are connected or not to the same food web and supported by similar basal resources. As mentioned

before, although this is out of the scope of this manuscript, these uncertainties, described briefly in the end of this paragraph and in the next, could have been disentangled by applying other biochemical tools such as CSIA-AA. Even with diet and baseline uncertainties, if the authors wish to display how the fractionation of $\delta^{15}\text{N}$ and $\delta^{66}\text{Zn}$ between *P. hispida* (or other prey species) and *U. maritimus* are related, the regression between predator-prey $\delta^{15}\text{N}$ and predator-prey $\delta^{66}\text{Zn}$ values could be added to the manuscript.

We thank the reviewer for their comment. We respectfully disagree with the first sentence of the reviewer here. As we write in our manuscript (original line 186) we are specifically talking about archaeological material here. For archaeological material, variability between a consumer's and a predator's $\delta^{15}\text{N}$ values independent of diet variability is a challenge for estimating a trophic level. If diet varied, that may also affect the trophic levels of the studied species (and challenge trophic level interpretations). Additionally, while we do discuss the diet variability in the supplements, we also note in the supplements and main text that for both species trophic level variability across the Arctic is likely low, as is today, i.e., even if prey items vary across sites to some degree, they likely still fed on a similar trophic level.

However, the reviewer raises an important point with the mobility of modern polar bears and ringed seals as this may influence $\Delta^{15}\text{N}_{U. maritimus - P. hispida}$ values. We discuss the mobility of both species now as a potential effect on $\Delta^{15}\text{N}_{U. maritimus - P. hispida}$ variability across sites, together with the citations provided by the reviewer (lines 329-333).

1.22 Line 253: Maybe I am missing something here, but what kind of bone was used in this research and how were they identified at species level?

We have added to supplementary table 1 a column with the information of which kind of bone was sampled. We also explain in more detail in the supplementary archaeological context how bones were identified. However, identifying bones in Arctic archaeological sites is quite straightforward, as mammalian biodiversity in the Arctic is extremely low and it is generally not very challenging to identify these taxa since there is only 2-3 taxa that a given bone could be from once you identify it as terrestrial or marine and large/medium/small.

1.23 Line 332: Did all the compared groups show a normal distribution and equal variances? In lines 122, 148 and 173 the paper states that variances were unequal between sites for $\delta^{15}\text{N}$ values of *P. hispida* and *U. maritimus*. What was the approach when the assumptions could not be met?

Before performing Anova analysis, all datasets ($\delta^{66}\text{Zn}$, $\delta^{15}\text{N}$ and $\delta^{13}\text{C}$) underwent both visual inspection and statistical tests to ensure normality and equal variance. However, in the case of $\delta^{15}\text{N}$ dataset specifically, visual inspection did not reveal unequal variance, but Levene's test did. At the time, we decided to perform Anova analysis nonetheless in order to avoid confusion by having different tests between the tracers ($\delta^{66}\text{Zn}$, $\delta^{15}\text{N}$ and $\delta^{13}\text{C}$). Additionally, there's a rule of thumb that ANOVA is robust to heterogeneity of variance so long as the largest variance is not more than 4 times the smallest variance. More specifically however, the Anova method controls the nominal type I error the best when all assumption is respected (which is the case for $\delta^{66}\text{Zn}$ and $\delta^{13}\text{C}$ datasets), so when deciding to use a single analysis like we did, using this method over a non-parametric option was easily justified.

However, and like the reviewer correctly pointed out, this created an awkward situation where the Levene's test results for the $\delta^{15}\text{N}$ dataset were first dismissed for the Anova but then later acknowledged. To correct this, we decided to use a different omnibus test for the $\delta^{15}\text{N}$ dataset that is specifically used when data violates the assumption of homogeneity of variances: the Welch Anova analysis. The following post-hoc test, Games-Howell, reveals similar results that with the ANOVA + Tukey test (47 statistically significant different pair-wise comparisons compared to 48). We nonetheless decided to keep to the Welch Anova + Games-Howell tests for the $\delta^{15}\text{N}$ dataset to avoid any and all confusion that would arise from performing the usual Anova to this dataset.

1.24 Figure 1: Please include latitude and longitude graticulates, scale bar and a North arrow to better indicate the sampling locations to the reader. This comment concern all the maps in the manuscript.

We anticipated readers to be generally familiar with most of the depicted locations. Still, we added latitude and longitude graticulates. As we now have latitude graticulates, a North arrow is redundant. We also did not include a scale bar, as we present a planar map, and the scale varies accordingly over the large area presented. We do indicate that it is a schematic map in the caption now to avoid confusion.

1.25 Table 1: Since there are only two samples of *D. leucas*, that correspond to the maximum and minimum isotope values, in my opinion the calculation of SD for that species is redundant.

Done. We deleted the SD for *D. leucas* samples.

Supplementary Material:

1.26 Line 160: I might have missed something, but at this point I have already gone through the whole manuscript but I still do not quite grasp the importance of the experiment described in the supplementary figure 1. The authors could describe this aspect of the work better in the introduction or in the supplementary introduction

Done. We now include some more information in the introduction (lines 122-125).

1.27 Line 179: This information should be part of the introduction, see comment above

Done. See response to comment above (1.26).

1.28 Line 189: I understand this might be out of scope for this manuscript, but what about other techniques that could also be applied to the collagen? Could CSIA-AA be also used in small samples?

CSIA-AA is not considered a method suitable for small samples, as it typically requires 3 mg of collagen material, which is 6 times more than is needed for bulk C and N isotopes and would require relatively large bone samples to extract this amount of collagen (typically at least 500 mg of bone). In contrast, $\delta^{66}\text{Zn}$ requires only 30 to 40 mg of bone, and in the case of Zn-rich marine mammals even less than 20 mg of bone material.

1.29 Line 194: Is this sentence related to the samples analysed in the present study? If so please specify

Yes. We now refer to supplementary table 1 to clarify.

1.30 Line 216: This goes back to my comment about how those bones were identified and what specific bones were used in the analysis.

We further elaborate on the identification of the bone fragments now. See our response to comment 1.22.

1.31 Line 270: Temporal variability in isotopic values of primary producers is also an issue that can be discussed here, since samples collected hundred and thousand years apart were analysed in the present study. Would be interesting to know how oceanic $\delta^{66}\text{Zn}$ baselines are expected to vary throughout this time frame and this can be discussed.

Again, we cannot discuss the variability in baseline $\delta^{66}\text{Zn}$ values in detail, as, due to the novelty of this method, no baseline $\delta^{66}\text{Zn}$ values have been measured yet. We do show in this study that for polar bears and ringed seals, $\delta^{66}\text{Zn}$ values can be directly compared between most sites, independent of age,

which contrasts $\delta^{15}\text{N}$ and $\delta^{13}\text{C}$. The latter are affected by local baseline variability. This implies low baseline variability between regions and through time. However, as the first study to touch upon this, we are limited as to how detailed we can discuss these observations without too much speculation. We anticipate the results of this study to motivate additional research into these issues, so that they may be discussed in greater detail in the future.

1.32 Line 280: This sentence needs rephrasing, as this is only true if all prey items are supported by the same basal organic matter source.

Done.

1.33 Line 283: Food webs have changed quite a lot in the past 1000 years, especially due to human interactions. Different ranges in isotopic values could be a results of those changes, producing more diverse, less overlapping food webs. See Saporiti et al 2014 “Longer and Less Overlapping Food Webs in Anthropogenically Disturbed Marine Ecosystems: Confirmations from the Past” PLoS One. Furthermore, ice cover can influence diet and ecology of both predator and prey species (Wing et al 2020 “Penguins and Seals Transport Limiting Nutrients Between Offshore Pelagic and Coastal Regions of Antarctica Under Changing Sea Ice” Ecosystems), which is known to have changed in the past millennia. Is this sentence still relevant in this context?

Yes. Because $\delta^{15}\text{N}$ values vary significantly between modern ringed seals from different locations, despite them likely feeding on similar trophic levels (i.e., highlighting baseline variability). Therefore $\delta^{15}\text{N}$ values may have also varied in the past without diet being the main control. Nevertheless, as we discuss these issues later in the Supplementary Discussion, we deleted this sentence here.

1.34 Line 329: I understand the advantages of $\delta^{66}\text{Zn}$ data, but as one of my earlier comments, nitrogen isotopic differences between those two species cannot be used as a trophic level estimates. Difficulties arise mainly from lack of baseline data from primary producers, diet, metabolism and consumption rate (prey availability) variability, for both consumer and predator.

We agree. Our intention here was to highlight the difficulties of defining the relative trophic level of two species towards each other, especially when comparing both across multiple locations. Meaning identifying that one species (e.g., polar bears) fed at one trophic level higher that the other (e.g., ringed seals). We placed this part now in the main text (following comment 2.2) and have also rephrased it accordingly.

1.35 Line 355: The information in this section should have been provided much earlier in the manuscript, in order to provide the basis for the trophic level estimates.

We have added information to the introduction (lines 111-116), the discussion (lines 284-303) and supplementary discussion (lines 450-455) to better explain how we use the relative trophic levels here. Also, we added a column to table 1 with estimated literature trophic levels.

1.36 Line 374: Are those results in line with the published literature?

Yes. We cite here also a publication (citation 11). We rephrased the sentence to clarify.

References

1. Whiteman, J. P., Smith, E. A. E., Besser, A. C. & Newsome, S. D. A guide to using compound-specific stable isotope analysis to study the fates of molecules in organisms and ecosystems. *Diversity* 11, 1–18 (2019).

2. Sabadel, A., Durante, L. & Wing, S. Stable isotopes of amino acids from reef fishes uncover Suess and nitrogen enrichment effects on local ecosystems. *Mar. Ecol. Prog. Ser.* 647, 149–160 (2020).

3. Evershed, R. P. et al. Compound-Specific Stable Isotope Analysis in Ecology and Paleoecology. in *Stable Isotopes in Ecology and Environmental Science* 480–540 (Blackwell Publishing Ltd, 2008). doi:10.1002/9780470691854.ch14

I congratulate the authors for the manuscript, which was a pleasure to read.

Leonardo Maia Durante

We thank Leonardo Maia Durante for his approval and very thorough and constructive feedback.

Reviewer #2 (Remarks to the Author):

2.1 The authors present $\delta^{66}\text{Zn}$, $\delta^{15}\text{N}$ and $\delta^{13}\text{C}$ values for four species of marine mammals from Arctic archaeological sites in order to assess how well the isotopic proxies perform against one another as trophic indicators. This builds on the work of Jaouen et al. 2016, expanding that analysis across many archaeological sites in the Arctic. The authors find that Zn is homogenous across for species across. A wide geographic range, whereas $\delta^{15}\text{N}$ is more variable, likely due to baseline effects. They suggest that $\delta^{66}\text{Zn}$ is a more reliable trophic proxy, but part of this analysis hinges on $\delta^{15}\text{N}$ derived trophic levels, which seems contradictory and needs further explanation.

We do not refer to $\delta^{66}\text{Zn}$ as a more or less reliable trophic level proxy. We suggest that, due to the lower baseline variability, $\delta^{66}\text{Zn}$ may provide a better inter-site comparability than $\delta^{15}\text{N}$ when investigating multiple species across multiple sites (original lines 176-177). This does not imply that $\delta^{15}\text{N}$ is an unreliable or less reliable trophic level proxy. In our main text, our interpretation of $\delta^{66}\text{Zn}$ values between species does not hinge on $\delta^{15}\text{N}$ derived trophic levels. Yes, $\delta^{15}\text{N}$ values vary between polar bears and ringed seals, but we do not use these values to justify our $\delta^{66}\text{Zn}$ results. Instead, we observe a difference in $\delta^{66}\text{Zn}$ values between predator (polar bear) and prey (ringed seal and others), which is the result of trophic Zn discrimination supported by literature references to $\delta^{66}\text{Zn}$ variability amongst organisms (e.g., Jaouen et al., 2016a, 2016b, Bourgon et al., 2020) and the known trophic interactions between polar bears and their prey species (e.g., Derocher et al., 2002, Hobson and Welch 1992, Hobson et al., 2002). The $\delta^{15}\text{N}$ values are used here for comparison. All trophic level estimates in the main text are relative trophic level estimates between species, with the exception of one mentioning of trophic levels calculated based on $\delta^{15}\text{N}$ values as calculated in the supplements to compare with the observed difference between polar bear and ringed seal $\delta^{66}\text{Zn}$ values, i.e., $\Delta^{66}\text{Zn}_{U. maritimus - P. hispida}$ values. We rephrased accordingly.

We realised, by comparing this comment with comment 1.16 that both reviewers read our use of $\delta^{66}\text{Zn}$ trophic levels contradictory to each other, with reviewer 1 stating: “there is no direct comparison between Zn isotopes and trophic levels of the species analysed in the main text.”. In order to clarify for both reviewers, we rephrased parts of the introduction (lines 111-116), the discussion (lines 284-303) and supplementary discussion (lines 450-455) to avoid further misinterpretation. We also added typical trophic levels from the literature to table 1. (see our response to 2.7).

We do estimate $\delta^{66}\text{Zn}$ trophic levels in the supplementary material based on $\delta^{15}\text{N}$ values in attempt to quantify $\delta^{66}\text{Zn}$ trophic level discrimination factors between polar bears and their prey species. However, we do indicate that these trophic level “estimations only represent oversimplified estimations, not considering population specific dietary differences, location specific baseline variations and organism specific trophic and tissue-type enrichment factors” (original Supplements line 363-365). We cannot calculate true $\delta^{66}\text{Zn}$ trophic level discrimination factors without knowing the exact diet, concentrations of Zn ingested, and dietary $\delta^{66}\text{Zn}$ composition. That is naturally beyond the scope of this study and also

impossible to achieve by analysing archaeological samples alone. Although speculative, the trophic level assessment is highly interesting given a $\delta^{66}\text{Zn}$ trophic level discrimination factor that appears to be relatively close to the “true” $\delta^{66}\text{Zn}$ trophic level discrimination factor as indicated by the comparison with terrestrial mammals and $\Delta^{66}\text{Zn}_{U. maritimus - P. hispida}$ values. For this reason, we included this section in the supplements. We have modified parts of this section now to avoid a similar confusion again (lines 450-455).

2.2 I think this is a great study, but I think it should be published in a longer format journal. All of this work with nontraditional isotopes is exciting and welcome contribution to the literature, but the authors need to clarify the discussion of their results, especially how $\delta^{15}\text{N}$ and $\delta^{66}\text{Zn}$ can be integrated when they claim that only one of those proxies is reliable. Also, a significant portion of the discussion and analysis have been bumped to the supplemental information, and I think this paper would be vastly improved by moving those points to the main text, especially the isotopic context section, which would have clarified many of the assumptions made in the main text. I also think the bone dissolution tests the authors did were incredibly useful, but it's hidden away in the supplement, as are the refined discussions on baseline isotopic values, preservation and diagenesis, non-dietary effects on trophic discrimination, etc.

We thank the reviewer for complimenting this study. We are sorry for the misunderstanding, but again, we never claim, that only one of the proxies is reliable. In fact, the opposite is the case: “As with $\delta^{15}\text{N}$ values, bone $\delta^{66}\text{Zn}$ values clearly demonstrate a trophic spacing between *U. maritimus* and *P. hispida* in all locations analysed” (original manuscript lines 169-170). In our manuscript we state that when it comes to comparing the same species across different Arctic sites, $\delta^{66}\text{Zn}$ values are more homogenous within a species than $\delta^{15}\text{N}$ values, likely due to a lower baseline variability. That is an advantage of Zn but does not imply $\delta^{15}\text{N}$ values are not reliable trophic level indicators.

We now include the isotopic context section as another introduction chapter in the main text (lines 126-189).

We also now include as suggested by the reviewer the section about non-dietary effects on trophic discrimination factors in the main text (lines 309-355).

While we agree that some of the information can or should be moved to the main text, we do not agree with moving all the supplementary information into the main text. By not including all the supplementary information, we also follow the 5000 word main text guideline. All the supplementary information we provide is important, yet the following sections of the supplements were not included in the main text with the reasoning behind our decision explained:

We have expanded on the bone dissolution experiments more in the introduction and results (lines 122-125 and 199-204, respectively). However, we want to keep the majority of the bone dissolution method results and discussion in the Supplements, as they are very technical and more methodological. We believe a too detailed discussion on these results would distract from the main message of the manuscript and break the flow of reading.

The refined discussion on preservation and diagenesis, while undoubtedly important, should in our opinion remain in the supplements, for the simple reason that it would due to the vast number of sites and samples analysed only distract from the more important findings of this study. This section deals with why we can perform this study. In the main text we want to focus on the study itself.

Likewise, we do not think that including the detailed discussion about the $\delta^{15}\text{N}$ and $\delta^{13}\text{C}$ baseline variability should be placed in the main text. It is a very detailed section, relying heavily on the cited literature and again distracting from the novel $\delta^{66}\text{Zn}$ results. However, after including the information from the isotopic context section to the main text, we agree with the reviewer that many parts of discussion are now clearer, especially those dealing with baseline variability. For those reasons, we believe that including this section now is not necessary and prefer to keep that more detailed section in the supplements for the more specialised readers.

We would also like to keep the trophic level assessment in the Supplements, for the simple reason that it is speculative and might lead to additional confusion as demonstrated by the comments of this revision.

We have added additional information from this section to the main text though to avoid this kind of confusion (lines 284-303). Please also see our response to comment 2.1. Additionally, we discuss in this section the $\delta^{66}\text{Zn}$ results from walrus bones analysed by Jaouen et al. (2016) in comparison to our data. While, again, we believe this to be valuable information, this part does not include new data from our study and is in part speculative. Still, we believe this part to inspire future $\delta^{66}\text{Zn}$ research dealing with these more specific issues, which is why we included this discussion in the supplements here for the more specialised readers.

In line with the suggestion of reviewer number 1, we now use section names when citing supplementary information to make it easier for the reader to find the information.

Figure comments:

2.3 The extended data figures are impossible to read. The Y axis has been compressed to the point that you can't accurately read values from it, and the stacks on stacks of lines showing significance among the data as the outcomes of the ANOVA are extraordinarily challenging to read with d15N and d13C data. All of these extended data figures should be made into tables.

Thank you for this comment. While our idea was to offer a way to visually illustrate the variability between the proxies (mainly through the sheer number of statistically significant different pair-wise comparisons), those graphs could indeed have been made in a clearer way. We accordingly decompressed the y-axis to allow more accurately reading and seeing the values for each site. Additionally, we have also added the pair-wise comparison results into tables and placed the figures into the supplementary information.

Results section

2.4 Four species of marine mammals are analyzed for C, N and Zn, but only the results of *P. hispida* are discussed in detail – why is that? Without thorough discussion of at least the other marine carnivore that was the focus of this study (*U. maritimus*), this is an incomplete section. It also seems that all figures for this section are only presented in the Supplement and not the main body of the text – there should be a citation for Figure 1 here as well.

We included additional information regarding *U. maritimus* isotope values and refer to figure 1. More thorough statistical analyses were performed on the *P. hispida* alone, as the sample size for other taxa did not allow to adequately investigate trends between sites any further. For example, the number of sites with *U. maritimus* samples is slightly smaller but most importantly, many sites contain significantly less individuals with some of the bordsens having only 2 individuals. The *U. maritimus* samples are thus mostly useful to assess the trophic spacing (and how homogenous/heterogenous it is) than their variability between site and across the Arctic. However, even without more thorough statistical analysis, the Levene's test adequately demonstrate that *P. hispida*'s situation seems to hold true for other taxa and this information is also reported in the results.

Discussion section

2.5 The authors lean heavily on the assumption that the base of Arctic food-web Zn must be more homogenous than for d13C and d15N in order to explain the lack of variation in Zn isotopes. This lack of variation is interesting, and definitely possible, but without direct measurements of $\delta^{66}\text{Zn}$ from the same plankton in this foodweb study, other possibilities must be discussed. It's possible (and also quite likely) that these higher carnivores have shifted prey sources and trophic positions through time, especially between the archaeological and modern samples. The authors should discuss this possibility in the text. They do discuss the varied diets of *P. hispida* and *U. maritimus* in the supplemental text. This discussion comes up later in the section (lines 186 – 199), but I think it should appear sooner for clarity.

We have been trying to identify other possibilities besides a more homogenous food web baseline, or a strong homogenisation of $\delta^{66}\text{Zn}$ values within lower trophic level organisms (also mentioned in the original text line 150). We discuss this in the main text. However, these are the only explanations for the observed $\delta^{66}\text{Zn}$ values in both polar bears and ringed seals. We demonstrate a trophic discrimination for $\delta^{66}\text{Zn}$ values between both species, also supported by a similar offset observed between terrestrial herbivores and carnivores (Jaouen et al., 2016; Bourgon et al., 2020) and a correlation of $\delta^{66}\text{Zn}$ with $\delta^{15}\text{N}$ when including all taxa. Therefore, if significant shifts in prey sources occurred (that also led to a shift in consumer trophic level) then we would also have to see this shift in $\delta^{66}\text{Zn}$ values. We find it highly unlikely that a shift in trophic level of these species among the sites would coincidentally lead to a homogenisation of $\delta^{66}\text{Zn}$ values within the same species and among sites (for 13 locations and 17 sites). Additionally, relative differences among sites in polar bear and ringed seal archaeological bones $\delta^{15}\text{N}$ and $\delta^{13}\text{C}$ values are comparable “in both spacing and amplitude with modern geographical variations observed from zooplankton^{26,27,28} and higher consumer soft tissue^{6,27,28,31} including *P. hispida*^{51,52,53,54,55,56} and *U. maritimus*⁴⁶” (original manuscript lines 136-138). This demonstrates that baseline is the main control on observed differences in $\delta^{15}\text{N}$ and $\delta^{13}\text{C}$ values between archaeological sites. Significant shifts in trophic level compared to today would have led to differences between archaeological and modern relative site differences in $\delta^{15}\text{N}$ and $\delta^{13}\text{C}$ values for a species not observed herein.

However, this comment did inspire us to include differences in dietary Zn uptake as a possibility for the statistical differences in $\delta^{66}\text{Zn}$ values for the QjJx-1 (Little Cornwallis Island), QkHn-13 (Devon Island) and SfFk-4 (eastern Ellesmere Island) based on post-hoc Tukey pair-wise comparison.

2.6 The authors state that $\delta^{66}\text{Zn}$ is more reliable than $\delta^{15}\text{N}$, but then use $\delta^{15}\text{N}$ to determine trophic level for these species in order to determine trophic level discrimination factors for Zn across the samples. This seems contradictory, and comes across as the authors using $\delta^{15}\text{N}$ only to validate the other isotope system, while writing off the $\delta^{15}\text{N}$ data as unreliable. This needs to be clarified and/or rewritten.

See also our response to 2.1. Again, we never once mention $\delta^{15}\text{N}$ as unreliable in the manuscript. We simply point out, that it is more effected by baseline variability than $\delta^{66}\text{Zn}$. We also do not use $\delta^{15}\text{N}$ to validate $\delta^{66}\text{Zn}$ trophic levels, when referring to differences between polar bear $\delta^{66}\text{Zn}$ values and their prey species. Only once do we estimate $\delta^{66}\text{Zn}$ discrimination factors based on $\delta^{15}\text{N}$, to compare with $\Delta^{66}\text{Zn}_{U. maritimus - P. hispida}$ values and terrestrial mammals in the supplementary discussion. We use $\delta^{15}\text{N}$ for that, as both reflect trophic levels as stated by us multiple times (original manuscript lines 169, 221-222, 240). We have now made this clearer in the introduction (lines 111-116), the discussion (lines 284-303) and supplementary discussion (lines 450-455) where we refer to the calculation based on $\delta^{15}\text{N}$.

2.7 I think something that would clarify this paper and make it easier to follow would be a summary table of trophic position estimates for all species in the study from both isotopic investigations (from this and previous studies) and also from the ecological literature. It is extremely challenging to follow the discussion because it's never very clear where inferred trophic position for each species is coming from, and whether or not that differs from previous research.

We added a summary with the range of trophic level estimates from the literature to table 1. We already have two tables in the supplements dealing with the trophic level estimates based on this study (supplementary table 7 and 8). Yet again (see our response to 2.1, 2.6) we typically refer to relative trophic levels in the main text, not absolute trophic levels.

2.8 Line 176: (and diet?) is a more casual writing expression, I would rewrite this to something more formal, eg, “...values for a specific taxon, and possibly diet...”.

Done.

Reviewer #3 (Remarks to the Author):

Review of “Advantages of zinc isotopes as a new dietary proxy in marine ecology” by McCormack et al.

3.1 This manuscript describes the utility of using ^{66}Zn isotopic compositions in archeological samples to determine relative trophic positions in the marine food web, in this case for polar bears and their main prey, seals. The main premise is that $\delta^{66}\text{Zn}$ values may be as useful or, in some cases, more useful than using $\delta^{15}\text{N}$ values as is traditionally done for ecological studies. This manuscript is of high quality, is well written and concise, and I don't see any flaws in the methodology, although I admit I am not an expert in non-traditional stable isotopes. I very much enjoyed reading it. It certainly may be a very useful technique for certain types of samples. I recommend it for publication either as is or subject to minor revision. Consequently, I only have a few comments for the authors as indicated below.

We thank the reviewer for his compliments and the approval of our work.

3.2 To me, the advantage of ^{66}Zn is use in archaeological samples where little is known about baseline or geographic variation in $\delta^{15}\text{N}$ values, or when collagen is absent. For “fresh” samples, $\delta^{15}\text{N}$ values should give the same information, but are much cheaper and far less labour intensive than the methods described for measurement of $\delta^{66}\text{Zn}$ values. This is not meant as a criticism, but rather I think the authors may want to emphasize that this method is ideally suited for archeological studies. As it is, the tone of the manuscript implies that this new method may be as useful as traditional stable isotope analyses for fresh samples. Calculated trophic levels and their relative error using $\delta^{66}\text{Zn}$ and $\delta^{15}\text{N}$ values are basically identical and indistinguishable (Supplementary Tables 7 and 8) so I think that for modern samples $\delta^{15}\text{N}$ analyses would be preferred.

We agree here with the reviewer that $\delta^{66}\text{Zn}$ is particularly useful in absence of collagen preservation, a topic of intensive research also from the authors of this study (e.g., Bourgon et al., 2020). However, we disagree with the notion that just because collagen is preserved bulk collagen $\delta^{15}\text{N}$ analysis should be performed instead of $\delta^{66}\text{Zn}$. Again, we never intended to imply that $\delta^{66}\text{Zn}$ should replace $\delta^{15}\text{N}$ in ecological studies. Instead, we argue repeatedly (Abstract, Discussion and Conclusion), that combining $\delta^{15}\text{N}$ with $\delta^{66}\text{Zn}$ will result in more robust (palaeo-) dietary interpretations. We highlight this even more now in the introduction (lines 54-56, 122-125) and by adding more information about the dissolution experiment (which should enable combining both methods) to the introduction and results as suggested by reviewer number 1 and 2.

3.3 The archaeological sites examined vary in age from 325-4100 y, with most samples spanning an age of 200 y or so. What effects would the authors expect from this? For instance, is it possible that the time span between individual samples (i.e. two samples from the same site could be up to 200 years apart in age) collectively serves to smooth out the differences between $\delta^{66}\text{Zn}$ values more so than $\delta^{15}\text{N}$ values of collagen? I'm fairly certain that the variation in baseline $\delta^{15}\text{N}$ or $\delta^{13}\text{C}$ values in the arctic over a period of 100's years is unknown.

This is a good point. We touch upon it in the discussion and the supplements but cannot discuss it in detail with the sample material at hand. We do not know how $\delta^{15}\text{N}$ and $\delta^{13}\text{C}$ baselines varied for all these Arctic sites over the timeframe analysed herein. However, as discussed in the Supplements 4.3 in detail, we see, very similar relative $\delta^{15}\text{N}$ and $\delta^{13}\text{C}$ patterns in modern POM and fauna (including seals and polar bears), compared to our archaeological material. We therefore conclude that baseline is the most important factor for the inter-site $\delta^{15}\text{N}$ and $\delta^{13}\text{C}$ variability within ringed seals and polar bears. The effect on Zn i.e., perhaps a smoothing out of baseline differences seems unlikely, as we compare multiple sites herein. Nevertheless, this is certainly an important point to be analysed in future studies, having the necessary sample material, which we do not have herein.

3.4 Title. Again, It is unclear from the title that this study deals with archeological samples and I suggest that this be included. At first glance it may be incorrectly assumed that this study deals with current (i.e. fresh) ecological samples.

Done. The title has been modified accordingly.

3.5 Introduction. l. 63. Baseline isotopic variations can also be compensated by compound specific amino acid isotopic analyses - see Webb et al, 2015 for an example application of this technique to collagen.

We added this information to the introduction (lines 42-45).

3.6 Extended Data Figures. Im not sure there is any way to graphically express the various p-values for the statistical tests, but these figures are certainly confusing at best. Im uncertain of the structure of the published article, but I would suggest that these be omitted from the main article and included in the supplementary material.

We modified these figures and added a table as suggested by reviewer 2, see our response to comment 2.3. We also placed the figures in the supplementary material.

Supplementary Information.

3.7 Baseline carbon and nitrogen isotope variability recored in P. Hispida bones. I suggest the authors further elaborate on possible reasons behind this variability. For example, terrestrial carbon influx from rivers, such as the McKenzie, may serve to drive down marine d13C values.

Done. We included this information in the supplements (lines 406-408).

3.8 Figure 1. Demineralized collagen intakt. Should be intact?

Done.

REVIEWERS' COMMENTS:

Reviewer #1 (Remarks to the Author):

The authors addressed all my comments clearly and concisely, making adjustment to the manuscript as needed. I think the manuscript has improved after the review process and it is in good shape for publication.

I congratulate the authors for the good work on this and I only have one more comment that I think should be addressed before publication:

Line 42-45: Although I think this sentence is well placed here, I think the end of the sentence "mostly applied to freshwater and terrestrial ecosystems" is misleading and should be changed. Almost all biogeochemical techniques are mostly applied to freshwater and terrestrial systems, because of the simple fact that there is more in-depth ecological research done in these systems. This is due to the better understanding of ecological process and easier access to research sites. Even though, there is a lot of CSIA-AA research done in marine systems:

1. Durante, L. M., Sabadel, A. J. M., Frew, R. D., Ingram, T. & Wing, S. R. Effects of fixatives on stable isotopes of fish muscle tissue: implications for trophic studies on preserved specimens. *Ecological Applications* 30, 1–16 (2020).
2. Eglite, E. et al. Strategies of amino acid supply in mesozooplankton during cyanobacteria blooms: A stable nitrogen isotope approach. *Ecosphere* 9, (2018).
3. Close, H. G. Compound-Specific Isotope Geochemistry in the Ocean. *Annual Review of Marine Science* 11, 27–56 (2019).
4. Whiteman, J. P., Smith, E. A. E., Besser, A. C. & Newsome, S. D. A guide to using compound-specific stable isotope analysis to study the fates of molecules in organisms and ecosystems. *Diversity* 11, 1–18 (2019).
5. Sabadel, A., Durante, L. & Wing, S. Stable isotopes of amino acids from reef fishes uncover Sues and nitrogen enrichment effects on local ecosystems. *Marine Ecology Progress Series* 647, 149–160 (2020).
6. Nielsen, J. M., Popp, B. N. & Winder, M. Meta-analysis of amino acid stable nitrogen isotope ratios for estimating trophic position in marine organisms. *Oecologia* 178, 631–642 (2015).

Leonardo Maia Durante

Reviewer #2 (Remarks to the Author):

I appreciate the author's extensive re-writes, which really clarify the paper. I especially like the addition of the isotopic context section. Overall, I'm impressed with these edits and I think the paper is ready for publication.

2.1 I appreciate the work the authors have done to rewrite and re-vamp this manuscript. Though the authors may not have intended to make the claim the Zn isotopes perform better than d15N, the tone of the paper had been set by this sentence in the abstract, which has now been removed: "d66Zn may more reliably record trophic levels between *U. maritimus* and prey species than d15N." I think this is why reviewer 3 and myself came away from this paper with that impression, and I'm glad the authors have reworked and rephrased the paper.

I still have issue with the fact that this kind of study was conducted on archaeological material instead of modern samples, where the authors themselves state that "We cannot

calculate true $\delta^{66}\text{Zn}$ trophic level discrimination factors without knowing the exact diet, concentrations of Zn ingested, and dietary $\delta^{66}\text{Zn}$ composition. That is naturally beyond the scope of this study and also impossible to achieve by analysing archaeological samples alone." There are also issues with shifting dietary patterns between archaeological and modern specimens, so making assumptions about trophic level will be inherently problematic. That being said, the authors layout this issues with more clarity than they did in the previous version.

2.2 I think my issues were adequately addressed with the re-writes. The clarity in navigating the supplemental info, referring too specific sections, has made the paper easier to read.

2.3 The figures are easier to read.

2.5 This was one of my biggest critiques of the paper and I think the authors have adequately addressed the comment.

2.6 The rewrites in the introduction have made this much clearer.

A couple notes:

Line 47: You could add "(palaeo)dietary" into this sentence to denote the use of the proxy for modern and fossil applications.

Line 68/69: "to what extent" instead of "to which extent"

Author response

We acknowledge the thorough and constructive feedback given by all reviewers.

In the following we discuss the comments made by each referee (original remarks are in black, our rebuttal/comments in blue).

REVIEWERS' COMMENTS:

Reviewer #1 (Remarks to the Author):

The authors addressed all my comments clearly and concisely, making adjustment to the manuscript as needed. I think the manuscript has improved after the review process and it is in good shape for publication.

I congratulate the authors for the good work on this and I only have one more comment that I think should be addressed before publication:

Line 42-45: Although I think this sentence is well placed here, I think the end of the sentence “mostly applied to freshwater and terrestrial ecosystems” is misleading and should be changed. Almost all biogeochemical techniques are mostly applied to freshwater and terrestrial systems, because of the simple fact that there is more in-depth ecological research done in these systems. This is due to the better understanding of ecological process and easier access to research sites. Even though, there is a lot of CSIA-AA research done in marine systems:

1. Durante, L. M., Sabadel, A. J. M., Frew, R. D., Ingram, T. & Wing, S. R. Effects of fixatives on stable isotopes of fish muscle tissue: implications for trophic studies on preserved specimens. *Ecological Applications* 30, 1–16 (2020).
2. Eglite, E. et al. Strategies of amino acid supply in mesozooplankton during cyanobacteria blooms: A stable nitrogen isotope approach. *Ecosphere* 9, (2018).
3. Close, H. G. Compound-Specific Isotope Geochemistry in the Ocean. *Annual Review of Marine Science* 11, 27–56 (2019).
4. Whiteman, J. P., Smith, E. A. E., Besser, A. C. & Newsome, S. D. A guide to using compound-specific stable isotope analysis to study the fates of molecules in organisms and ecosystems. *Diversity* 11, 1–18 (2019).
5. Sabadel, A., Durante, L. & Wing, S. Stable isotopes of amino acids from reef fishes uncover Suess and nitrogen enrichment effects on local ecosystems. *Marine Ecology Progress Series* 647, 149–160 (2020).
6. Nielsen, J. M., Popp, B. N. & Winder, M. Meta-analysis of amino acid stable nitrogen isotope ratios for estimating trophic position in marine organisms. *Oecologia* 178, 631–642 (2015).

Leonardo Maia Durante

We thank Leonardo Maia Durante for his constructive feedback and his approval of our work. We have deleted “mostly applied to freshwater and terrestrial ecosystems” as requested.

Reviewer #2 (Remarks to the Author):

I appreciate the author's extensive re-writes, which really clarify the paper. I especially like the addition of the isotopic context section. Overall, I'm impressed with these edits and I think the paper is ready for publication.

We thank the reviewer for his approval.

2.1 I appreciate the work the authors have done to rewrite and re-vamp this manuscript. Though the authors may not have intended to make the claim the Zn isotopes perform better than d15N, the tone of the paper had been set by this sentence in the abstract, which has now been removed: "d66Zn may more reliably record trophic levels between *U. maritimus* and prey species than d15N." I think this is why reviewer 3 and myself came away from this paper with that impression, and I'm glad the authors have reworked and rephrased the paper.

I still have issue with the fact that this kind of study was conducted on archaeological material instead of modern samples, where the authors themselves state that "We cannot calculate true d66Zn trophic level discrimination factors without knowing the exact diet, concentrations of Zn ingested, and dietary d66Zn composition. That is naturally beyond the scope of this study and also impossible to achieve by analysing archaeological samples alone." There are also issues with shifting dietary patterns between archaeological and modern specimens, so making assumptions about trophic level will be inherently problematic. That being said, the authors layout this issues with more clarity than they did in the previous version.

We thank the reviewer for his comment and are glad, that our changes made our intensions clearer. We selected archaeological material, as this allowed us to sample this increasable large amount of samples, as well as investigate the use of our methods for both ecologists and archaeologists. As pointed out by the reviewer, we state all limitations. However, if we had sampled modern material, we would have faced the same issue, as we would still not have baseline information, which would have required sampling of POM across the Arctic, which is naturally not possible in this context.

2.2 I think my issues were adequately addressed with the re-writes. The clarity in navigating the supplemental info, referring too specific sections, has made the paper easier to read.

2.3 The figures are easier to read.

2.5 This was one of my biggest critiques of the paper and I think the authors have adequately addressed the comment.

2.6 The rewrites in the introduction have made this much clearer.

2.2 to 2.6: We thank the reviewer for his comments.

A couple notes:

Line 47: You could add "(palaeo)dietary" into this sentence to denote the use of the proxy for modern and fossil applications.

Done.

Line 68/69: "to what extent" instead of "to which extent"

Done.